# Separating common from salient patterns with Contrastive Representation Learning

**Robin Louiset**[1,2*] **Edouard Duchesnay**[1] **Antoine Grigis**[1] **Pietro Gori**[2]

## Abstract

Contrastive Analysis is a sub-field of Representation Learning that aims at separating common factors of variation between two datasets, a background (i.e., healthy subjects) and a target (i.e., diseased subjects), from the salient factors of variation, only present in the target dataset. Despite their relevance, current models based on Variational Auto-Encoders have shown poor performance in learning semantically-expressive representations. On the other hand, Contrastive Representation Learning has shown tremendous performance leaps in various applications (classification, clustering, etc.). In this work, we propose to leverage the ability of Contrastive Learning to learn semantically expressive representations well adapted for Contrastive Analysis. We reformulate it under the lens of the InfoMax Principle and identify two Mutual Information terms to maximize and one to minimize. We decompose the first two terms into an Alignment and a Uniformity term, as commonly done in Contrastive Learning. Then, we motivate a novel Mutual Information minimization strategy to prevent information leakage between common and salient distributions. We validate our method, called SepCLR, on three visual datasets and three medical datasets, specifically conceived to assess the pattern separation capability in Contrastive Analysis. Code available at `https://github.com/neurospin-projects/2024_rlouiset_sep_clr`

## 1 Introduction

In Representation Learning, practitioners estimate parametric models tailored to learn meaningful and compact representations from high-dimensional data. The objective is to capture relevant features to facilitate downstream tasks such as classification, clustering, segmentation, or generation. Contrastive Representation Learning (CL) has made remarkable progress in learning representations that encode high-level semantic information about inputs such as images (Zbontar et al. (2021); Wei et al. (2020); Bachman et al. (2019); He et al. (2020); Goyal et al. (2021); Dufumier et al. (2023); Barbano et al. (2023a)) and sequential data (Oord et al. (2019); Kong et al. (2019); Tian et al. (2020); Schneider et al. (2023); Sun et al. (2019)). With a distinct perspective, Contrastive Analysis (CA) approaches aim to discover the underlying generative factors that 1) distinguish a target dataset from a background dataset (*i.e.,* salient factors) and that 2) are shared between them (*i.e.,* common factors). It is usually assumed that target samples comprise additional (or modified) patterns compared to background samples (Abid et al. (2018); Zou et al. (2013; 2022); Severson et al. (2018); Ruiz et al. (2019); Ge & Zou (2016); Li et al. (2021); Zou et al. (2023)). The ability to *distinguish* and *separate* common from salient generative factors is crucial in various domains. For instance, in medical imaging, researchers seek to identify pathological patterns in a population of patients (target) compared to healthy controls (background) Antelmi et al. (2019); Aglinskas et al. (2022). Contrastive Analysis also concerns other domains like drug research (medicated vs. placebo populations), surgery (pre-intervention vs. post-intervention groups), time series (signal vs. signal-free samples), biology and genetics (control vs. characteristic-trait population, Jones et al. (2021) ).
Current Contrastive Analysis methods are based on VAEs (Variational Auto-Encoders) Kingma & Welling (2014). This choice is particularly suitable for generation and image-level manipulations. However, as shown in Phuong et al. (2018), VAE can fail to learn meaningful latent representations, or even learn trivial representations when the decoder is too powerful Chen et al. (2017). Conversely, Contrastive Learning (CL) methods have demonstrated outstanding results in many domains, such

---

*Corresponding author: `robin.louiset@gmail.com`
[1] NeuroSpin, University Paris Saclay, France. [2] LTCI, Télécom Paris, IPParis, France

as unsupervised learning Chen et al. (2020), deep clustering Li et al. (2020), content vs style identification von Kügelgen et al. (2021), background debiasing Wang et al. (2022); Ding et al. (2022), and multi-modality Yuan et al. (2021). This performance gap might be explained by the following reasons. CL methods produce representations invariant to a set of user-defined image transformations (translation, zoom, color jittering, etc.), whereas VAEs are highly sensitive to these uninteresting variability factors. Furthermore, VAEs maximize the log-likelihood, which is only a function of the marginal distribution of the input data and *not* of the latent representations. Differently, CL methods based on the InfoNCE loss implicitly maximize the Mutual Information (MI) between input data and latent features[1]. From a representation learning point of view, this makes much sense since the MI depends on the joint distribution between input data and representation Zhao et al. (2019). Inspired by these works, we propose to reformulate the Contrastive Analysis problem under the lens of the well-known InfoMax principle Bell & Sejnowski (1995); Hjelm et al. (2019) and leverage the representation power of Contrastive Learning (CL) to estimate the MI terms of our newly proposed Contrastive Analysis setting. We seek to separate the *salient patterns* of the target dataset from the *shared (common) patterns* with the background dataset. Common factors $c$ should be representative of both target and background datasets (respectively $y$ and $x$). Thus, we propose to maximize the MI between $x$ (resp. $y$) and $c$. We compute the Entropy and Alignment to estimate the MI, as in Wang & Isola (2020) and Rodríguez-Gálvez et al. (2023). Since salient factors $s$ should only describe patterns typical of the target data $y$, we propose to maximize the MI between $s$ and **only** $y$. Furthermore, we also add the constraint that background samples' representations should always be equal to an informationless vector $s'$ in the salient space. This objective is close to other recent CA ideas, such as in Contrastive PCA Abid & Zou (2019) and CA-VAEs, but also to Supervised Anomaly Detection intuitions, such as in DeepSAD Ruff et al. (2019), where the entropy of anomalies (eq. target) is maximized, whereas normal samples (eq. backgrounds) are set to a constant vector. We propose an extension of this salient term when fine-grained target attributes are available and propose disentangling these attributes within the salient space in a supervised manner. Moreover, to avoid information leakage between the common $c$ and salient space $s$, we constrain the MI to be (exactly) equal to 0. This choice avoids undesirable results since minimizing the MI may bring to a trivial solution where $c$ and/or $s$ would contain no information. Instead, we propose a method to estimate and maximize their joint entropy $H(c, s)$ without requiring any assumptions about the form of its pdf nor a neural network-based approximation. Our contributions are summarized below:

1. We introduce SepCLR, a novel theoretical framework for Contrastive Analysis based on the InfoMax principle. We identify three Mutual Information terms: a common space term, a salient space term, and a common-salient independence term.

2. We leverage Contrastive Learning to estimate the common and salient terms. We show how usual contrastive losses such as InfoNCE and SupCLR can be retrieved from the InfoMax Principle. Likewise, we derive a novel contrastive method to capture target-specific variability while canceling background variability in the salient space.

3. To reduce the information leakage between the common and salient spaces, we suggest a strategy that overcomes the pitfalls of usual Mutual Information Minimization methods.

## 2 RELATED WORKS

Our work relates to contrastive learning, mutual information, and contrastive analysis.

**Contrastive Learning and the InfoMax Principle.** Contrastive Learning (CL) hinges on an intuition that dates back to Becker & Hinton (1992). Given an input sample $x$ (image or sequence) and two different views (*i.e.,* transformations) $v$ and $v^+$ of $x$ that potentially overlap (spatially or sequentially), CL is based on the assumption that $v$ and $v^+$ should share a similar information content. A parametric encoder $f_\theta$ is then estimated by maximizing their "agreement" in the representation space so that their similarity/dependence is preserved in the embeddings $f_\theta(v)$ and $f_\theta(v^+)$. A commonly used measure of agreement is the Mutual Information between the two views embeddings that is maximized: $\theta^* \leftarrow \arg\max I(f_\theta(v); f_\theta(v^+))$, where the choice of $f_\theta$ imposes some structural constraints (*i.e.,* inductive bias). As shown in Tschannen et al. (2019), this objective can be seen as a lower bound on the InfoMax principle $\max_\theta I(x; f_\theta(x))$ (Linsker (1988), Bell & Sejnowski (1995)).

---

[1]as shown in Sec.3.1, the MI between the latent representation of two views, maximized in many recent methods, is a lower bound of the MI between input data and latent representations.

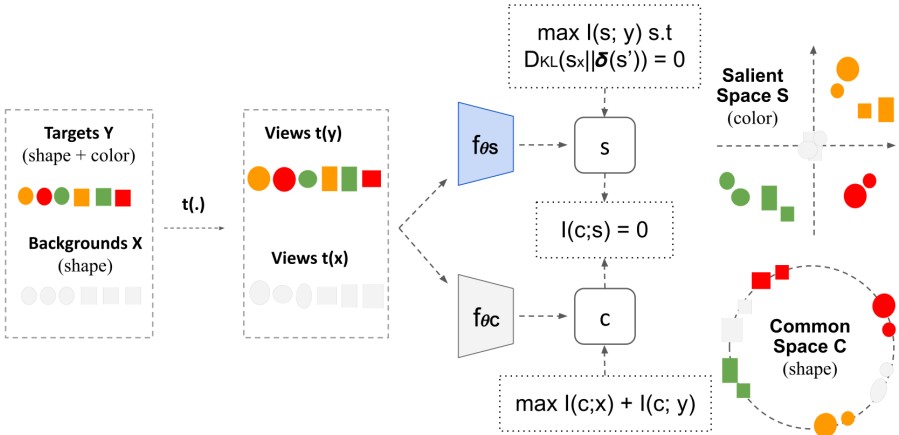

Figure 1: SepCLR is trained to identify and separate the salient patterns (color variations) of the target dataset $Y$ from the common patterns (shape) shared between background $X$ and target dataset $Y$. Views (transformations $t(\cdot)$) of both datasets are fed to two different encoders, one for the salient space ($f_{\theta_s}$) and one for the common space ($f_{\theta_c}$). In the hyperspherical common space, $C$, embeddings of views of the same image (from both $X$ and $Y$) are aligned, while embeddings from different images are repelled (max $I(c;x)+I(c;y)$). This enforces $C$ to represent the shared patterns (shape). In the salient space $S$, which is a Euclidean space, in order not to capture background variability (*i.e:* shape), background embeddings are aligned onto an information-less null vector **s'** ($D_{KL}(s_x||\delta(s')) = 0$). Furthermore, embeddings of views of the same image (only from $Y$) are aligned while embeddings from different images are pushed away from each other, and they are all repelled from **s'** (max $I(s;y)$). This enforces $S$ to capture only the salient patterns of $Y$ (color). To limit the information leakage between $C$ and $S$, their MI is constrained to be null, *i.e:* $I(c;s=0)$.

Many approaches (Kong et al. (2019); Tian et al. (2020); Bachman et al. (2019); Oord et al. (2019); Tsai et al. (2020); Barbano et al. (2023b)) propose to maximize $I(f_\theta(v); f_\theta(v^+))$ rather than the original InfoMax objective since the embeddings $f(x)$ have a lower dimension than the original samples $x$ and the choice of the transformation for the views gives more flexibility. Wang & Isola (2020) simplifies the usual CL loss InfoNCE Chen et al. (2020) into an alignment (or reconstruction) and a uniformity (or entropy) term. While the alignment term trains the encoder to assign similar representations to positive views, the uniformity term encourages feature distribution to preserve maximal information *i.e.:* maximal entropy. Recently, Rodríguez-Gálvez et al. (2023) demonstrated that these terms could be derived from the maximization of $I(f_\theta(V); f_\theta(V^+))$ and that several clustering methods could be retrieved from this formulation. We build onto these works to introduce the CL framework required to develop the proposed CA losses.

**Contrastive Analysis.** Contrastive Analysis (CA) methods are designed to separate salient latent variables (*i.e:* patterns that are specific to the target dataset) from common latent variables (*i.e:* patterns that are shared between background and target datasets). Recently, contrastive Variational Auto-Encoders were designed to capture higher-level semantics Abid & Zou (2019); Zou et al. (2022); Weinberger et al. (2022); Louiset et al. (2023). These methods usually rely on a latent space split into two parts, common and salient, estimated by two different encoders. To limit information leakage between common and salient spaces, three types of regularization have been proposed. First, a usual solution is to introduce an explicit regularization on the salient encoder to minimize the background information expressivity Abid & Zou (2019); Weinberger et al. (2022); Louiset et al. (2023); Zou et al. (2022; 2023). This regularization forces the salient vectors of the backgrounds to be close to **s'** (information-less vector, often equal to 0). A second idea, proposed in MM-cVAE Weinberger et al. (2022), is to match the common space distributions of the target and background samples by minimizing their Maximum Mean Discrepancy (MMD) Gretton et al. (2012). This regularization reduces the information that would enable discriminating targets from background samples within the common space. In cVAE Abid & Zou (2019), and SepVAE Louiset et al. (2023), authors minimize the Mutual Information between the common and salient spaces.

**Contrastive Analysis is not disentanglement nor style vs. content separation.** CA is not about disentanglement, which aims to isolate independent variation factors in a single data-set Locatello et al. (2020); Chen et al. (2018); N et al. (2017). In contrast, CA seeks to separate common from

target-specific generative factors without requiring the isolation of independent factors usually defined in a supervised manner using external attributes. Furthermore, CA is not about separating style from content (Kazemi et al. (2019); von Kügelgen et al. (2021)), where content is usually defined as the invariant part of the latent space, namely the part shared across different views. In contrast, style refers to the varying part that accounts for the differences between views. Content and style depend on the chosen semantic-invariant transformations, and they are defined for a single dataset. In CA, we do not necessarily need transformations or views, and we jointly analyze two different datasets.

**Mutual Information Minimization.** Mutual Information minimization has gained significant attention in diverse applications such as disentangling Kim & Mnih (2018); N et al. (2017), domain adaptation Gholami et al. (2018), style/content identification Kazemi et al. (2019), and Information Bottleneck compression Alemi (2020). Typically, it can serve as a regularizer to diminish the dependence between variables. However, computing the value of Mutual Information is hardly possible in cases where closed forms of density functions, joint or marginal, are unknown. In most machine learning setups, access is limited to only samples drawn from the joint distribution. To accommodate, most estimation methods (lower bound, upper bound, and reliable estimators) focus on sample-based estimation. However, most of these works either require strong assumptions about one of the distributions (*e.g.,* its form) (Alemi (2020); Poole et al. (2019)) or the introduction of an independent neural network to approximate a distribution in a sample-based variational manner. For instance, CLUB Cheng et al. (2020) derives an upper bound of the Mutual Information $I(X, Y)$ by either assuming the closed-form of $p(y|x)$ or, in its variational form, estimating it with a parameterized neural network $q_\theta(y|x)$. Another example concerns Total Correlation methods Louiset et al. (2023); Kim & Mnih (2018) that leverage the Density Ratio trick Sugiyama et al. (2012); Nguyen et al. (2010) to estimate the density ratio between the joint distribution and the product of the marginals. This technique demands optimizing an independent discriminator to discriminate samples drawn from the joint distribution from those drawn from the product of the marginals.

## 3 THE INFOMAX PRINCIPLE FOR CONTRASTIVE ANALYSIS

Let $X = \{x_i\}_{i=1}^{N_X}$ and $Y = \{y_j\}_{j=1}^{N_Y}$ be the background and target data-sets of images respectively. As it is commonly done in Contrastive Analysis Abid & Zou (2019); Weinberger et al. (2022); Louiset et al. (2023), we suppose that both $x_i$ and $y_j$ are drawn i.i.d. from the *same* conditional distribution $p_\theta(\cdot|c, s)$, that is parameterized by unknown parameters $\theta$ and that depends on two latent variables: the **common** generative factors $c \in \mathbf{R}^{D_c}$, shared between $X$ and $Y$, and the **salient** (or **target-specific**) generative factors $s \in \mathbf{R}^{D_s}$, which are only present in $Y$ and not in $X$. The separation between $c$ and $s$ can be considered a weakly supervised learning problem since the only level of supervision is the population-based label $X$ or $Y$. The user has no knowledge about the common and salient generative factors at training (or test) time. By grounding our method on the InfoMax principle Bell & Sejnowski (1995); Hjelm et al. (2019), and since we want the common factors $c$ to be representative of both datasets, we propose to maximize the mutual information $I$ between $c$ and both datasets $X$ and $Y$. Similarly, we propose maximizing the mutual information between the salient factors $s$ and **only** the target samples $Y$. Since we want the background samples $x$ to be fully encoded by $c$, we enforce the salient factors $s$ of $x$ to be always equal to a constant value $s'$ (*i.e.,* no information): $x_i \sim p_\theta(x|c_i, s_i = s')$. Mathematically, we do that by minimizing the Kullback–Leibler divergence $D_{KL}$ between $p(s|x)$ and $\delta(s')$, a Dirac Delta distribution centered at $s'$. Furthermore, to enforce the separation (*i.e.,* independence) between $c$ and $s$, we also propose to use $I(c, s) = 0$ as a regularization constraint.

Our objective is to *separate* and *infer* the common $c$ and salient $s$ factors given the input data $X$ and $Y$. We use two probabilistic encoders, $f_{\theta_c}$ and $f_{\theta_s}$, parameterised by $\theta_c$ and $\theta_s$, to approximate the conditional distributions $p(c|\cdot)$ and $p(s|\cdot)$ respectively. The two encoders are shared between $X$ and $Y$. Furthermore, as commonly done in recent representation learning papers, we assume to have multiple views $v$ of each image $x$ (or $y$) generated via a stochastic augmentation function $t$: $v = t(\cdot)$. By denoting $c = f_{\theta_c}(v)$, $s = f_{\theta_s}(v)$, $s_x = f_{\theta_s}(t(x))$, our goal becomes finding the optimal parameters $\theta^* = \{\theta_c^*, \theta_s^*\}$ that maximize the following cost function:

$$\arg\max_\theta \underbrace{\lambda_C(I(x; c) + I(y; c))}_{\text{Common InfoMax}} + \underbrace{\lambda_S I(y; s)}_{\text{Salient InfoMax}} \quad \text{s.t.} \underbrace{D_{KL}(s_x||\delta(s')) = 0}_{\text{Information-less hyp.}} \text{ and } \underbrace{I(c, s) = 0}_{\text{Independence hyp.}} \quad (1)$$

In Sec. 3.1, we show how to estimate the common terms, $I(x; c)$ and $I(y; c)$, via a formulation similar to the alignment and entropy terms introduced in Wang & Isola (2020). In Sec. 3.2, we take into account the information-less hypothesis (*i.e.* background embeddings should always be equal to an information-less vector in the salient space) to estimate the salient term $I(y; s)$. Ultimately, in Sec. 3.3, we propose a strategy to enforce the independence hypothesis *i.e.* $I(c; s) = 0$, that prevents information leakage between the common and salient space.

## 3.1 RETRIEVE INFONCE FROM INFOMAX FOR COMMON SPACE

In this section, we demonstrate that $I(x; c)$ and $I(y; c)$ can be estimated via the multi-view alignment and uniformity losses inspired by Wang & Isola (2020). Full derivation can be found in Appendix Sec. A. Let $f_{\theta_C}$ be the common encoder and $c \sim f_{\theta_C}(t(.))$ be the common representations. The MI $I(x; c)$ (same reasoning is also valid for $I(y; c)$) can be decomposed into:

$$I(x; c) = \underbrace{-\mathbf{E}_{x \sim p_x} H(c|x)}_{\text{Alignment}} + \underbrace{H(c)}_{\text{Entropy}} \tag{2}$$

**Entropy (Uniformity).** As in Wang & Isola (2020), the entropy can be computed with a non-parametric estimator described in Ahmad & Lin (1976). To do so, we compute the approximate density function $\hat{p}(c_i)$ with a Kernel Density Estimator as in Parzen (1962); Rosenblatt (1956), based on views $v_j$ (random augmentation of an image with index $j$) uniformly sampled from both the target dataset $f_{\theta_c}(t(y)) \sim p(c|y)$ and the background dataset $f_{\theta_c}(t(x)) \sim p(c|x)$. We choose a Gaussian kernel with constant standard deviation $\tau$, which results in an L2 distance between the views. However, in practice, we constrain the outputs $f_{\theta_C}(.)$ to be unit-normed, which is equivalent to directly choosing a von Mises-Fischer kernel with concentration parameter $\frac{1}{\tau}$. [2] As in Wang & Isola (2020), we optimize a lower bound of this estimator in practice, called $-\mathcal{L}_{\text{unif}}$:

$$\mathcal{L}_{\text{unif}} = \log \frac{1}{N_X + N_Y} \sum_{i=1}^{N_X + N_Y} \frac{1}{N_X + N_Y} \sum_{j=1}^{N_X + N_Y} \exp \frac{-||f_{\theta_C}(v_i) - f_{\theta_C}(v_j)||_2^2}{2\tau} + \underbrace{\log \sqrt{2\pi\tau}}_{\text{Constant term}} \tag{3}$$

**Alignment:** Differently from Wang & Isola (2020), we propose to estimate the conditional entropy $-H(c|x)$ with a re-substitution entropy estimator. We compute the approximate density function $\hat{p}(c_i|x_i)$ with a Kernel Density Estimator based on samples uniformly drawn from the conditional distribution $c_i^k \sim p(c|x_i)$, where $c_i^k = f_\theta(v_i^k)$ and $v_i^k$ are $K$ views obtained via the stochastic process $t(.)$. As for the entropy term, we choose a Gaussian kernel with constant standard deviation $\tau$ to derive an L2 distance between the views. Our formulation generalizes Wang & Isola (2020), as we directly retrieve a multi-view alignment term between $K$ positive views of the same image and not a single-view alignment as in Wang & Isola (2020). However, in practice, to reduce the computational burden, we also choose a single view K=1, as in Wang & Isola (2020). Combining the background alignment $-H(c|x)$ and the target alignment $-H(c|y)$, we obtain:

$$\mathcal{L}_{\text{align}} = -\frac{1}{N_X + N_Y} \sum_{i=1}^{N_X + N_Y} \log \frac{1}{K} \sum_{k=1}^{K} \exp \frac{-||f_{\theta_C}(v_i) - f_{\theta_C}(v_i^k)||_2^2}{2\tau} + \underbrace{\log(\sqrt{2\pi\tau})}_{\text{Constant term}} \tag{4}$$

**On the relation with $I(\mathbf{f}_\theta(\mathbf{v}), \mathbf{f}_\theta(\mathbf{v}'))$.** Many recent representation learning works (Chen & He (2020); Wang & Isola (2020)) maximize the MI between two views $v$ and $v'$ of $x$: $I(f_\theta(v), f_\theta(v'))$. Inspired by the InfoMax principle, we propose instead maximizing $I(f_\theta(v), x)$. As shown in Tschannen et al. (2019), by directly applying the *data processing inequality*, one can demonstrate that $I(f_\theta(v), f_\theta(v'))$ is a lower bound of $I(f_\theta(v), x)$.

## 3.2 DERIVE THE BACKGROUND-CONTRASTING ALIGNMENT AND UNIFORMITY TERMS

In this section, we consider the maximization of the salient term $I(s; y)$, which is decomposed into an alignment and uniformity term as before, constrained by the information-less hypothesis:

$$\arg\max I(s; y) = -\underbrace{\mathbf{E}_{y \sim p_y} H(s|y)}_{\text{Target-only Alignment}} + \underbrace{H(s)}_{s'\text{-Entropy}} \quad \text{s.t.} \quad \underbrace{D_{KL}(s_x || \delta(s')) = 0}_{\text{Information-less hyp.}} \tag{5}$$

---

[2] Intuitively, if $||f_{\theta_C}(.)||_2 = 1$, the L2 distance between two representations can be simplified into a negative dot-product: $||f_{\theta_C}(v_i) - f_{\theta_C}(v_j)||_2 = 2 - 2 f_{\theta_C}(v_i)^T . f_{\theta_C}(v_j)$. Full proof in Appendix Sec .A.2.1.

**Target-only alignment.** To estimate the target samples' alignment term, we use the same estimation method as in 3.1. Namely, we derive an alignment term between two views $v_i = t(y_i)$ and $v_i^k = t(y_i^k)$ of the same target image $y_i$. As in Sec. 3.1, we use a re-substitution estimator with a Gaussian Kernel Density Estimation with constant standard deviation $\tau$ and $K = 1$ in practice.

$$\mathcal{L}_{\text{y-align}} = -\frac{1}{N_Y} \sum_{i=1}^{N_Y} \log \frac{1}{K} \sum_{k=1}^{K} \exp \frac{-||f_{\theta_S}(v_i) - f_{\theta_S}(v_i^k)||_2^2}{2\tau} + \underbrace{\log(\sqrt{2\pi\tau})}_{\text{Constant term}} \tag{6}$$

$s'$**-Uniformity.** Concerning the Entropy term, as in Sec. 3.1, we propose to develop the salient entropy with a re-substitution entropy estimator. Again, we use a lower bound of $\hat{H}(S)$ called $-\mathcal{L}_{s'\text{-unif}}$. Then, we estimate the density $\hat{p}(s)$ with a Gaussian Kernel Density Estimator based on samples uniformly drawn from the target dataset $f_{\theta_S}(t(y)) \sim p(s|y)$ and from the background dataset $f_{\theta_S}(t(x)) \sim p(s|x)$. Importantly, the information-less hypothesis constrains the salient encoder to produce background embeddings always equal to the information-less vector $s'$: $f_{\theta_S}(t(x)) \sim \delta(s')$. Using this hypothesis in the computations (see Sec. B.2 of the Supplementary) and ignoring constant terms, we obtain:

$$\mathcal{L}_{s'\text{-unif}} = \log \frac{1}{N_Y} \sum_{i=1}^{N_Y} \left( \exp \frac{-||f_{\theta_s}(t(y_i)) - s'||_2^2}{\tau} + \frac{1}{N_Y} \sum_{j=1}^{N_Y} \exp \frac{-||f_{\theta_s}(t(y_i)) - f_{\theta_s}(t(y_j))||_2^2}{2\tau} \right) \tag{7}$$

To respect the Information-less hypothesis, we re-write Eq. 5 as a Lagrangian function, with the constraint expressed as a $\beta$-weighted ($\beta \geq 0$) KL regularization. Assuming that $s_x$ follows a Gaussian distribution centered on $f_{\theta_s}(x)$ with a standard deviation $\tau$ (constant hyper-parameter), we derive the KL divergence as an L2-distance between $f_{\theta_s}(t(x))$ and $s'$, as in He et al. (2019):

$$\mathcal{F}(\theta_S, \beta; x, y, s) = -\lambda_S \mathcal{L}_{\text{y-align}} - \lambda_S \mathcal{L}_{s'\text{-unif}} - \beta \frac{1}{N_X} \sum_{i=1}^{N_X} \frac{||f_{\theta_s}(t(x_i)) - s'||_2^2}{2\tau} \tag{8}$$

## 3.3 ON THE NULL MUTUAL INFORMATION CONSTRAINT

In Eq. 1, to avoid information leakage between common and salient space, we constrain our problem so that the MI between $c$ and $s$ is null. Another common choice would be to simply *minimize* $I(c, s)$ instead than forcing it to be equal to 0. In Tab. 1 and 2, we show that the latter choice (*i.e.,* $I(c, s) = 0$) clearly outperforms (variational) MI minimization methods, as vCLUB Cheng et al. (2020), vL1out Poole et al. (2019), vUB Alemi (2020), and TC Louiset et al. (2023), (see Sec. F.7). **Minimizing $H(c) + H(s)$ is detrimental:** By def., $I(c; s) = -H(c, s) + H(c) + H(s) \geq 0$, which entails $H(c; s) \leq H(c) + H(s)$. Thus, a trivial way to minimize $I(c; s)$ would be minimizing $H(c) + H(s)$. However, it reduces the quantity of information contained in either $c$ or $s$, which could be detrimental. Furthermore, the Common and Salient InfoMax losses of our framework seek to maximize $H(c)$ and $H(s)$ rather than minimizing them. This is why, instead of minimizing $I(c; s)$, we propose to simultaneously maximize $H(c, s)$, $H(c)$ and $H(s)$, until $H(c, s) = H(c) + H(s)$, to respect the constraint $I(c; s) = 0$.[3]

To estimate and maximize $H(c, s)$, we propose a new method, called **kernel-based Joint Entropy Maximization (k-JEM)**, that requires no assumptions about the form of the pdf nor a neural network-based approximation (Cheng et al. (2020); Alemi (2020); Poole et al. (2019)). Inspired by Holmes et al. (2007), we develop $H(c, s)$ with a re-substitution entropy estimator: $-\hat{H}(c, s) = \frac{1}{N_X + N_Y} \sum_{i=1}^{N_X + N_Y} \log \hat{p}(c_i, s_i)$. We estimate the density $\hat{p}_\theta(c_i, s_i)$ with a Gaussian Kernel Density Estimation with a constant standard deviation parameter $\tau$ with samples $(c, s)$ uniformly drawn from the target dataset $(f_{\theta_S}(t(y)), f_{\theta_C}(t(y))) \sim p(c, s|y)$ and from the background dataset $(f_{\theta_S}(t(x)), f_{\theta_C}(t(x))) \sim p(c, s|x)$. The indices $i$ and $j$ refer to two different samples in the dataset. Full computations in Appendix, Sec. D.

$$-H(c, s) = \frac{1}{N_X + N_Y} \sum_{i=1}^{N_X + N_Y} \log \frac{1}{N_X + N_Y} \sum_{j=1}^{N} \exp \frac{-||c_i - c_j||_2^2}{2\tau} \exp \frac{-||s_i - s_j||_2^2}{2\tau} \tag{9}$$

---

[3]In this work, we implicitly assume that the encoders $f_{\theta_c}$ and $f_{\theta_s}$ can model any distribution.

## 4 DISENTANGLING ATTRIBUTES IN THE SALIENT SPACE

Here, we propose to explore an extension of the salient contrastive loss in the case where independent fine-grained attributes about the target dataset $\{a_i \in \mathcal{R}^{D_S}\}_{i=1}^{N_Y}$ are available. We assume the existence of $D_S$ attributes and each attribute $a^d$ is generated by a single factor of generation $s_y^d$ of the target dataset. We also make the hypothesis that the given attributes describe the entire salient variability of the target dataset,[4] and thus construct our salient encoder to output (exactly) $D_S$ latent dimensions. We aim to construct a salient space where each salient latent dimension $S^{d_s}$ only depends on its corresponding attribute $a^{d_s}$. By leveraging the attributes in a supervised manner, we re-write Eq. 1 by replacing $I(y; s)$ with the sum of all attribute Supervised InfoMax terms :

$$\arg \max I(x; c) + I(y; c) + \frac{1}{D_S} \sum_{d=1}^{D_s} \underbrace{I(a^d; s^d)}_{\text{d-th SupInfoMax}} \qquad \text{s.t. } D_{KL}(s_x||\delta(s')) = 0 \text{ and } I(c, s) = 0 \quad (10)$$

Taking inspiration from Dufumier et al. (2021a;b), we then decompose each d-th attribute Supervised InfoMax term in a supervised alignment and uniformity term:

$$I(a^d; s^d) \geq \frac{1}{N_Y} \sum_{i=1}^{N_Y} w_\sigma(a_i^d, a_j^d) \frac{||s_i^d - s_j^d||_2}{\tau} + \hat{H}(s^d) = \mathcal{L}_{\text{d-th SupInfoMax}} \quad (11)$$

where the indices $i$ and $j$ refer to two different samples in the target dataset and the scalar weight $w_\sigma(a_i^d, a_j^d) = \frac{K_A(a_i^d, a_j^d)}{\sum_{j=1} K_A(a_i^d, a_j^d)}$ measures the similarity between their attributes. We define $K_A$ as a Gaussian kernel and the entropy $\hat{H}(s^d)$ is also estimated, as before, with a Gaussian kernel.

## 5 EXPERIMENTS

Here, we measure our method's ability to separate common from target-specific variability factors. We train a Logistic (or Linear) Regression on inferred factors to assess whether the information about a characteristic present in both populations, or only in the target one, is captured in the common (C) or in the salient (S) latent space. We compute (Balanced) Accuracy scores (=(B-)ACC), or Area-under Curve scores (=AUC) for categorical variables, Mean Average Error (=MAE) for continuous variables, and the sum of the differences ($\delta_{tot}$) between the obtained results and the expected ones.

**Hyper-parameters.** We empirically choose $\tau = 0.5$ for all experiments and losses. The other hyper-parameters are $\lambda_C$ and $\lambda_S$, which weigh the common terms and salient terms, respectively, and $\lambda$, which weighs the independence regularization. The choice of these weights depends on the ratio between common and target-salient information quantity, which might differ among datasets. Architectures and hyper-parameters are chosen as the top-performing ones for each experiment.

**SOTA CA methods** We have compared the performance of our method with the most recent and best-performing CA-VAE methods whose code was available: cVAE Abid & Zou (2019) [5], conVAE Aglinskas et al. (2022) [6], MM-cVAE Weinberger et al. (2022) and SepVAE Louiset et al. (2023). In each experiment, all CA-VAE use the same encoder-decoder architecture, as described in the Supplementary F. The architecture used for SepCLR is also described in Sec. F.

**Digits superimposed on natural backgrounds.** In this experiment particularly suited to CA and inspired from Zou et al. (2013), we consider CIFAR-10 images as the background dataset ($y = 0$) and CIFAR-10 images with an overlaid digit as the target dataset ($y = 1$). In Tab .1, our model outperforms all other methods in correctly capturing the common factors of variability (*i.e:* objects) in the common space and the target-specific factors of variability (*i.e:* digits) in the salient space.

**CelebA accessories dataset.** We consider a subset of CelebA Liu et al.. The target class ($y = 1$) contains images of celebrities wearing glasses or hats. The background class ($y = 0$) contains images of celebrities without accessories. In Tab .2, SepCLR correctly captures the information that enables distinguishing glasses from hats only in the salient space, and it puts the information to distinguish men from females in the common space. Our method globally outperforms all other methods (smallest $\delta_{tot}$). "Best Expected" reports perfect results (100%) when the attribute should be present in that latent space and a random result (50%) when it should not.

---

[4]If it is not true, one can add a Salient InfoMax term (Eq.1) and increase the salient space dimension.

[5]Here, for cVAE, we use the fixed version of the TC regularization described in Louiset et al. (2023).

[6]Here, conVAE corresponds to cVAE method without the TC regularization, as in Aglinskas et al. (2022).

Table 1: Digits on CIFAR-10 (B-ACC). Details in Sec.F.7. Table 2: CelebA accessories (B-ACC).

| | DIGITS | | OBJECTS | | $\delta_{\text{TOT}}\downarrow$ | HATS/GLSS | | SEX | | $\delta_{\text{TOT}}\downarrow$ |
|---|---|---|---|---|---|---|---|---|---|---|
| | S↑ | C↓ | S↓ | C↑ | | S↑ | C↓ | S↓ | C↑ | |
| cVAE | 90.6 | 23.0 | 11.2 | 33.4 | 90.2 | 83.89 | 66.56 | 60.25 | 60.60 | 82.32 |
| ConVAE | 86.2 | 21.0 | 10.6 | 35.6 | 89.8 | 81.64 | 65.94 | 61.53 | 58.93 | 86.90 |
| MM-cVAE | 88.8 | 19.6 | 12.2 | 32.0 | 93.6 | 84.60 | 66.43 | 60.56 | 61.57 | 80.82 |
| SepVAE | 90.6 | 17.8 | 10.6 | 36.6 | 81.2 | 84.46 | 65.19 | 60.12 | 59.20 | 81.65 |
| SepCLR-vCLUB SYM | 94.4 | 18.0 | 8.0 | 14.6 | 97.0 | 98.98 | 59.62 | 65.20 | 54.23 | 71.61 |
| SepCLR-vCLUB C → S | 95.2 | 39.4 | 9.2 | 27.2 | 106.2 | 98.81 | 73.71 | 61.77 | 53.72 | 82.95 |
| SepCLR-vCLUB S → C | 95.2 | 57.0 | 8.8 | 31.8 | 118.8 | 98.66 | 95.95 | 67.65 | 73.16 | 91.78 |
| SepCLR-vL1o SYM | 95.0 | 18.4 | 8.4 | 15.4 | 96.4 | 98.83 | 56.94 | 57.97 | 51.38 | 64.60 |
| SepCLR-vL1o C → S | 94.0 | 23.0 | 10.0 | 31.8 | 87.2 | 99.04 | 93.17 | 63.35 | 59.13 | 89.91 |
| SepCLR-vL1o S → C | 95.4 | 41.0 | 9.2 | 28.8 | 106.0 | 98.46 | 94.33 | 65.00 | 71.77 | 89.10 |
| SepCLR-vUB SYM | 94.6 | 42.0 | 8.2 | 29.0 | 106.6 | 98.68 | 87.33 | 63.59 | 56.09 | 96.15 |
| SepCLR-vUB C → S | 92.8 | 23.4 | 7.8 | 22.6 | 95.8 | 98.73 | 94.40 | 66.58 | 71.37 | 90.88 |
| SepCLR-vUB S → C | 96.6 | 41.8 | 8.6 | 28.6 | 105.2 | 98.78 | 93.92 | 62.94 | 61.27 | 96.81 |
| SepCLR-TC | 95.2 | 68.6 | 10.2 | 24.2 | 139.4 | 98.97 | 98.76 | 60.39 | 74.96 | 85.22 |
| SepCLR-MMD | 94.6 | 21.2 | 9.0 | 62.2 | 53.4 | 98.95 | 67.50 | 71.83 | 65.47 | 74.91 |
| SepCLR-NO K-JEM | 95.6 | 94.4 | 9.0 | 42.0 | 145.8 | 99.03 | 66.68 | 98.48 | 79.48 | 86.65 |
| SepCLR-K-MI | 96.2 | 19.8 | 8.0 | 65.8 | 45.8 | 98.96 | 77.10 | 63.07 | 71.08 | 70.13 |
| SepCLR-K-JEM | 96.2 | 11.0 | 10.4 | 73.2 | **32.0** | 98.57 | 55.21 | 62.52 | 78.00 | **41.16** |
| BEST EXPECTED | 100.0 | 10.0 | 10.0 | 100.0 | 0.0 | 100.0 | 50.0 | 50.0 | 100.0 | 0.0 |

**Neuroimaging: parsing schizophrenia's heterogeneity.** Separating healthy from pathological latent mechanisms that drive neuro-anatomical variability in schizophrenia is challenging. Yet, this ability could help understand and anticipate the development of these diseases. Given healthy MRI scans and patients with schizophrenia, we aim to capture pathological patterns only in the salient space that should correlate with clinical scales (such as positive symptoms: SAPS, and negative symptoms: SANS) while not being biased by demographic variables (age, sex or acquisition sites), which should be encoded in the common space. As in Louiset et al. (2021; 2023), we gather T1w VBM Ashburner (2000) warped MRIs ($128^3$ voxels) and evaluate our method in a cross-validation scheme. In Table 3, we can clearly see that our method outperforms all others.

Table 3: Separate healthy from pathological variability in Schizophrenia disorder. Best in **bold**.

| | AGE MAE | | SEX B-ACC | | SITE B-ACC | |
|---|---|---|---|---|---|---|
| | C↓ | S↑ | C↑ | S↓ | C↑ | S↓ |
| cVAE | 6.43±0.18 | 7.27±0.25 | 75.06±3.48 | 74.99±2.15 | 65.12±4.06 | 59.62±5.42 |
| ConVAE | 6.40±0.26 | 7.46±0.18 | 74.45±1.80 | 72.72±1.32 | 60.42±3.67 | 54.46±2.46 |
| MM-cVAE | 6.55±0.18 | 7.10±0.34 | 72.80±3.95 | 72.15±2.47 | 63.24±1.41 | 56.69±9.84 |
| SepVAE | **6.40±0.13** | **7.98±0.25** | 74.19±1.81 | 72.61±2.19 | 63.89±2.16 | 44.10±5.78 |
| SepCLR-K-JEM | 6.64±0.21 | 7.72±0.45 | **76.5±1.98** | **70.85±1.89** | **66.94±5.06** | **42.40±4.91** |
| | SANS MAE | | SAPS MAE | | DIAGNOSIS | |
| | C↑ | S↓ | C↑ | S↓ | C↓ | S↑ |
| cVAE | 5.89±0.67 | 4.35±0.26 | 4.65±0.34 | 2.98±0.18 | 60.66±2.63 | 68.24±5.42 |
| ConVAE | 6.17±0.45 | 3.95±0.28 | 4.50±0.37 | 2.76±0.18 | 61.85±2.60 | 58.53±4.87 |
| MM-cVAE | 6.78±0.54 | 4.92±0.58 | 4.52±0.33 | 3.16±0.05 | 64.25±2.98 | 70.94±4.08 |
| SepVAE | 7.05±0.67 | 4.14±0.39 | 4.79±0.67 | 2.60±0.27 | 60.90±1.75 | 79.15±3.39 |
| SepCLR-K-JEM | **9.17±2.49** | **3.74±0.12** | **5.54±0.70** | **2.52±0.16** | **60.16±1.19** | **79.90±1.57** |

**Chest and eye pathologies subtyping.** We propose two experiments using subsets of CheXpert Irvin (2019) and ODIR dataset (Ocular Disease Intelligent Recognition dataset) [7] to assess the ability of our method in a controlled environment. About CheXpert, we have healthy X-ray scans (background) and diseased scans (target) divided into 3 distinct subtypes: cardiomegaly, lung edema, and pleural effusion. In the ODIR dataset, there are healthy (background) and diseased fundus images (target) which are divided into 5 subtypes: Diabetes, Glaucoma, Cataract, Age macular degeneration, and pathological Myopia. Sex-related patterns should only be captured in the common encoder.
**Disentangling dSprites while contrasting with a background.** To evaluate CA method enriched with target attributes, we provide a novel toy dataset. Background dataset $X$ consists of 4 MNIST digits (1-4) regularly placed on a grid. Target dataset $Y$ consists of a dSprites item added upon the grid of digits. dSprites only exhibit 5 generation factors (shape, zoom, rotation, X position, Y position). Using Eq. 10, we train our salient encoders in a supervised manner to capture and disentangle each attribute in a single salient space dimension (Fig. 2a). The common encoder is instead trained to capture the background variability (Fig. 2b). Quantitatively, 1st salient dimension distinguishes shapes (B-ACC= $98.23\%$) while the concatenation of other salient dimensions and common dimensions does not (B-ACC= $36.08\%$). $2_{\text{nd}}$ predicts zoom attribute ($R^2 = 0.977$) while others don't

---

[7]https://www.kaggle.com/datasets/andrewmvd/ocular-disease-recognition-odir5k

0.002. $3_{\text{rd}}$ predicts rotation ($R^2 = 0.947$), others don't $R^2 = 0.0$. $4_{\text{th}}$ predicts horizontal translation ($R^2 = 0.995$), others don't $R^2 = 0.017$. $5_{\text{th}}$ predicts vertical translation ($R^2 = 0.995$), others don't $R^2 = 0.009$. This shows that our method correctly *separate* common from salient information and *disentangle* salient factors (in a supervised manner) *at the same time*.

Table 4: CheXpert X-ray scans (B-ACC).      Table 5: ODIR fundus images (B-ACC).

| | SUBTYPE | | SEX | | $\delta_{\text{TOT}} \downarrow$ | SUBTYPE | | SEX | | $\delta_{\text{TOT}} \downarrow$ |
|---|---|---|---|---|---|---|---|---|---|---|
| | S $\uparrow$ | C $\downarrow$ | S $\downarrow$ | C $\uparrow$ | | S $\uparrow$ | C $\downarrow$ | S $\downarrow$ | C $\uparrow$ | |
| CVAE | 45.77 | 49.27 | 54.24 | 81.26 | 93.48 | 46.13 | 43.91 | 49.11 | 51.86 | 120.03 |
| CONVAE | 42.31 | 52.53 | 60.88 | 79.30 | 108.8 | 49.80 | 52.01 | 50.82 | 47.01 | 131.86 |
| MM-CVAE | 42.50 | 50.89 | 57.04 | 80.19 | 102.24 | 42.79 | 43.66 | 54.91 | 53.76 | 131.02 |
| SEPVAE | 42.20 | 51.10 | 56.38 | 79.95 | 102.34 | 38.64 | 41.44 | 52.91 | 52.62 | 124.75 |
| SEPCLR-K-JEM | 61.30 | 52.85 | 61.57 | 80.25 | **89.87** | 68.54 | 47.71 | 52.48 | 59.62 | **97.03** |
| BEST EXPECTED | 100.0 | 33.0 | 50.0 | 100.0 | 0.0 | 100.0 | 25.0 | 50.0 | 100.0 | 0.0 |

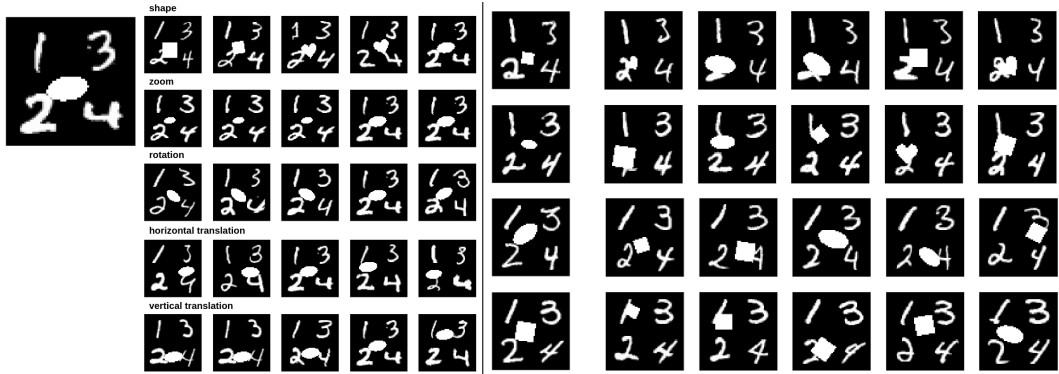

(a) Given the latent vector of the upper left image, we modify only one salient dimension in each row while freezing the others, then fetch the image in the dataset whose latent vector is the closest.

(b) Given the common latent vector of an image (left column), we fetch the image in the dataset whose inferred common latent vector is the closest in terms of L2 distance.

Figure 2: Qualitative results on attribute-supervised SepCLR

## 6 LIMITATIONS AND PERSPECTIVES

An important question in Contrastive Analysis, is the identifiability of the models. Namely, under which conditions can the models recover the true latent factors of the underlying data-generating process. Recent works have shown that non-linear models, VAEs included, are generally not identifiable. To obtain identifiability, two different solutions have been proposed: 1) either regularizing Kivva et al. (2022) the encoder or 2) introducing an auxiliary variable so that the latent factors are conditionally independent given the auxiliary variable Hyvarinen et al. (2019); Khemakhem et al. (2020). In CA, neither of these solutions may be used [8]. Even though SepCLR effectively *separates* common from salient factors, it does not assure that *all* true generative factors have been identified (like all CA methods). This is a serious limitation of CA methods that we leave for future works. Intriguingly, we also noticed that adding a reconstruction loss during the training degrades performance, see Sec. G.2 in Appendix. However, adding a powerful generator, as in Zou et al. (2023); Carton et al. (2024), on top of the frozen encoders would allow synthesizing new images and increase interpretability.

## 7 CONCLUSION

In this paper, we leverage the power of Contrastive Learning to learn semantically relevant representations for Contrastive Analysis. We reformulate Contrastive Analysis as a constrained InfoMax paradigm. Then, we propose to estimate the Mutual Information terms via alignment and uniformity terms. Importantly, we motivate a novel independence term between common and salient spaces computed via Kernel Density Estimation (KDE). Our method outperforms related works on toy, natural, and medical datasets specifically made to evaluate the common/salient separation ability.

---

[8]The dataset label could be considered as an auxiliary variable, but it does not make $c$ and $s$ independent

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

## A    RETRIEVE THE INFONCE LOSS

Let $X = \{(x_i)\}_{i=1}^{N}$ be the data-set of background images $x_i$, and $Y = \{(x_i)\}_{i=1}^{N}$ be the data-set of target images $y_i$. Input samples are assumed to be independently generated from latent unobserved variables $C = \{c_i \in \mathbf{R}^D\}_{i=1}^{N}$. Our objective is to estimate an encoder $f_{\theta_C}$ that infers latent factors of generation $c$ from the inputs (and its views $v$).

To do so, we entitle $c$ the latent codes produced by the common encoder $f_{\theta_C}(t(.))$, where $t(.) = v$ are the views generated from either $x$ or $y$ via a stochastic augmentation function $t(.)$. The objective is to construct an encoder $f_{\theta_C}(t(.))$ that is invariant to data augmentation. From the InfoMax perspective, we seek the optimal parameters $\theta^*$ that maximize the MI between $x$ and $c \sim f_{\theta_C}(t(x))$. Foremost, we decompose the MI $I(x; c)$ into:

$$I(x; c) = \underbrace{-\mathbf{E}_{x \sim p_x} H(c|x)}_{\text{Alignment}} + \underbrace{H(c)}_{\text{Entropy}} \tag{12}$$

but the same reasoning is valid for the target dataset: $I(c; y) = -\mathbf{E}_{y \sim p_y} H(c|y) + H(c)$.

### A.1    DERIVE THE UNIFORMITY TERM FROM THE ENTROPY TERM

In this section, we propose to make the correspondence between the concept of Entropy, well-known in Mutual Information literature, and the concept of Uniformity introduced in Wang & Isola (2020). The entropy can be derived with a non-parametric estimator described in Ahmad & Lin (1976) with samples uniformly drawn from both the target dataset and the background dataset.

$$\hat{H}(c) = -\frac{1}{N_X + N_Y} \sum_{i=1}^{N_X + N_Y} \log \hat{p}(c_i) \tag{13}$$

Then, we compute the approximate density function $\hat{p}(c_i)$ with a Kernel Density Estimator, based on samples uniformly drawn from both the target dataset $f_{\theta_c}(t(y)) \sim p(c|y)$ and the background dataset $f_{\theta_c}(t(x)) \sim p(c|x)$:

$$\hat{H}(c) = -\frac{1}{N_X + N_Y} \sum_{i=1}^{N_X + N_Y} \log \frac{1}{N_X + N_Y} \sum_{j=1}^{N_X + N_Y} K_C(c_i, c_j) \tag{14}$$

For simplicity, we choose a Gaussian kernel with constant standard deviation $\tau$ to derive an L2 distance between the views. This enables us to obtain:

$$\hat{H}(c) = -\frac{1}{N_X + N_Y} \sum_{i=1}^{N_X + N_Y} \log \frac{1}{N_X + N_Y} \sum_{j=1}^{N} \exp \frac{-||f_\theta(v_i) - f_\theta(v_j)||_2^2}{2\tau} + \underbrace{\log(\sqrt{2\pi\tau})}_{\text{Constant term}} \tag{15}$$

where $c_j = f_\theta(v_j)$ and $c_i = f_\theta(v_i)$. And where $v_i$ and $v_j$ are the views obtained by feeding the input with index $i$ (can be a target or a background sample) through the stochastic data augmentation function $t(.)$. In practice, Wang & Isola (2020) minimize the asymptotic lower bound of this term entitled Uniformity term. Using Jensen's inequality, we obtain:

$$\underbrace{-\frac{1}{N_X + N_Y} \sum_{i=1}^{N_X + N_Y} \log \frac{1}{N_X + N_Y} \sum_{j=1}^{N_X + N_Y} \exp \frac{-||f_\theta(v_i) - f_\theta(v_j)||_2^2}{2\tau}}_{= \hat{H}(c) - \log(\sqrt{2\pi\tau})} \geq$$
$$\underbrace{-\log \frac{1}{N_X + N_Y} \sum_{i=1}^{N_X + N_Y} \frac{1}{N_X + N_Y} \sum_{j=1}^{N_X + N_Y} \exp \frac{-||f_{\theta_C}(v_i) - f_{\theta_C}(v_j)||_2^2}{2\tau}}_{= -\mathcal{L}_{\text{uniform}}} \tag{16}$$

Given a bounded support, minimizing $\mathcal{L}_{\text{unif}}$ encourages the latent vectors to match a uniform distribution (*e.g:* spherical uniform distribution on unit-norm support in Wang & Isola (2020)).

## A.2 Derive the Multi-View Alignment term

Differently from Wang & Isola (2020), we propose to estimate the conditional entropy on background samples $-H(c|x)$ with a re-substitution entropy estimator.

$$-H(c|x) = \frac{1}{N_X} \sum_{i=1}^{N_X} \log \hat{p}(c_i|x_i) \tag{17}$$

We compute the approximate density function $\hat{p}(c_i|x_i)$ with a Kernel Density Estimator based on samples uniformly drawn from the conditional distribution $c_i^k \sim p(c|x_i)$, where $c_i^k = f_\theta(v_i^k)$ and $v_i^k$ are $K$ views obtained via the stochastic process $t(.)$.

$$-H(c|x) = \frac{1}{N_X} \sum_{i=1}^{N_X} \log \frac{1}{K} \sum_{k=1}^{K} K_C(f_{\theta_C}(v_i)), f_{\theta_C}(v_i^k)) \tag{18}$$

$K_C$ is chosen as a von Mises-Fisher kernel with a constant concentration parameter $\kappa = \frac{1}{\tau}$. These choices enable us to retrieve a Multi-View Alignment term with $K$ positive views rather than only 1 as in Wang & Isola (2020):

$$-H(c|x) + \log(C(\kappa)) = \frac{1}{N_X} \sum_{i=1}^{N_X} \log \frac{1}{K} \sum_{k=1}^{K} \exp \frac{-||f_{\theta_C}(v_i) - f_{\theta_C}(v_i^k)||_2^2}{2\tau} \tag{19}$$

By estimating the conditional entropy on target samples $-H(c|y)$ in the same fashion and summing both, we can retrieve the Alignment term written in Eq. 4. For computational reasons, we restrict to only one view in this paper: $K = 1$.

### A.2.1 On the connection between the Gaussian kernel and the von Mises-Fisher kernel

Let us note the kernel similarity between two representations: $f_{\theta_C}(x_i)$ and $f_{\theta_C}(x_j)$ as $K(f_{\theta_C}(x_i), f_{\theta_C}(x_j))$. Assuming that we are given a Gaussian kernel with a constant standard deviation $\sigma$, this term can be estimated as:

$$K_{\text{Gaussian}}(f_{\theta_C}(x_i), f_{\theta_C}(x_j)) = \frac{1}{\sqrt{2\pi\tau}} \exp \frac{-||f_{\theta_C}(x_i) - f_{\theta_C}(x_j)||_2^2}{2\tau} \tag{20}$$

Now, we can divide the square norm into three terms:

$$K_{\text{Gaussian}}(f_{\theta_C}(x_i), f_{\theta_C}(x_j)) = \frac{1}{\sqrt{2\pi\tau}} \exp \frac{-||f_{\theta_C}(x_i)||_2^2 - 2f_{\theta_C}(x_i)^T . f_{\theta_C}(x_j) + ||f_{\theta_C}(x_j)||_2^2}{2\tau} \tag{21}$$

Let assume that $f_{\theta_C}(x_i)$ and $f_{\theta_C}(x_j)$ are unit-normed, then this estimation get simplified into:

$$K_{\text{Gaussian}}(f_{\theta_C}(x_i), f_{\theta_C}(x_j)) = \frac{1}{\sqrt{2\pi\tau}} \exp \frac{-1 + f_{\theta_C}(x_i)^T . f_{\theta_C}(x_j)}{\tau} \tag{22}$$

which can be further simplified:

$$K_{\text{Gaussian}}(f_{\theta_C}(x_i), f_{\theta_C}(x_j)) = \frac{1}{e^1 \sqrt{2\pi\tau}} \exp \frac{f_{\theta_C}(x_i)^T . f_{\theta_C}(x_j)}{\tau} \tag{23}$$

Ignoring the normalization terms, we recognize the von Mises-Fisher kernel with concentration hyper-parameter $\kappa = \frac{1}{\tau}$:

$$K_{\text{vMF}} = \frac{1}{C(\kappa)} \exp \frac{f_{\theta_C}(x_i)^T . f_{\theta_C}(x_j)}{\tau}$$

## B Derive the Background-Contrasting InfoNCE loss in the salient space

In this section, we propose deriving the salient term $I(s; y)$ into a novel loss entitled BC-InfoNCE. Foremost, let us decompose the constrained Mutual Information maximization:

$$\arg\max - \underbrace{\mathbf{E}_{y \sim p_y} H(s|y)}_{\text{Target Alignment}} + \underbrace{H(s)}_{s'\text{-Entropy}} \quad \text{s.t.} \quad \underbrace{D_{KL}(s_x||\delta(s')) = 0}_{\text{Information-less hyp.}} \tag{24}$$

### B.1 ALIGNMENT OF TARGET SAMPLES:

In order to estimate the target samples' alignment term, we use the same estimation method as in 3.1. First, we derive an alignment term between two views $f_{\theta_S}(v_i)$ and $f_{\theta_S}(v_i^+)$ of the same target image $y_i$ using re-substitution estimation:

$$-\mathbf{E}_{y \sim p_y} \hat{H}(s|y) = \frac{1}{N_Y} \sum_{i=1}^{N_Y} \hat{p}(s_i|y_i) \tag{25}$$

Then, the density $\hat{p}(s_i|y_i)$ is estimated with a Kernel Density Estimator based on samples uniformly drawn from $p_{s|y_i}$, i.e.: $\{f_\theta(t(y_i)^k)|y_i\}_{k=1}^K$, where $t(y_i)^k = v_i^k$ are $K$ views uniformly drawn from the stochastic input-transformation process $t(y_i)$:

$$-\mathbf{E}_{y \sim p_y} \hat{H}(s|y) = \frac{1}{N} \sum_{i=1}^{N} \log \frac{1}{N} \sum_{k=1}^{K} K_Z(f_\theta(v_i), f_\theta(v_i^k)) \tag{26}$$

$K_Z$ is chosen as a von Mises-Fisher kernel with a constant concentration parameter $\kappa = \frac{1}{\tau}$ and only $K = 1$ positive view is chosen. These choices enable us to derive the target alignment term:

$$-\mathbf{E}_{y \sim p_y} \hat{H}(s|y) = \frac{1}{N_Y} \sum_{i=1}^{N_Y} \frac{-||f_\theta(v_i) - f_\theta(v_i^+)||_2^2}{2\tau} \tag{27}$$

### B.2 $s'$-UNIFORMITY:

Now, concerning the Entropy term, we propose to develop the salient entropy with a resubstitution entropy estimator from samples drawn from $X \cup Y$.

$$\hat{H}(S) = -\frac{1}{(N_Y + N_X)} \sum_{v \in t(X \cup Y)} \log \hat{p}(f_{\theta_S}(v)) \tag{28}$$

Then we estimate the density $\hat{p}(f_{\theta_S}(v))$ with a Gaussian Kernel Density Estimator based on latent vectors drawn from the target view $f_{\theta_S}(t(y))$ and from the background views $f_{\theta_S}(t(x))$.

$$\hat{H}(s) = -\frac{1}{N_Y + N_X} \sum_{v \in t(X \cup Y)} \log \frac{1}{N_Y + N_X} \sum_{v^+ \in t(X \cup Y)} \exp \frac{-||f_{\theta_s}(v) - f_{\theta_s}(v^+)||_2^2}{\tau} \tag{29}$$

We consider the asymptotic form of the Entropy. Therefore, we pull the $\log$ out of the exterior sum. In practice, it is equivalent to considering a lower bound of the Entropy. Now, separating the background and the target datasets inside the $\log$ yields:

$$
\begin{aligned}
\exp \mathcal{L}_{s'\text{uniform}} = & -\frac{1}{N_Y + N_X} \sum_{i=1}^{N_Y} \frac{1}{N_Y + N_X} \sum_{j=1}^{N_X} \exp \frac{-||f_{\theta_s}(y_i) - f_{\theta_s}(x_j)||_2^2}{\tau} \\
& -\frac{1}{N_Y + N_X} \sum_{i=1}^{N_Y} \frac{1}{N_Y + N_X} \sum_{j=1}^{N_Y} \exp \frac{-||f_{\theta_s}(y_i) - f_{\theta_s}(y_j)||_2^2}{\tau} \\
& -\frac{1}{N_Y + N_X} \sum_{i=1}^{N_X} \frac{1}{N_Y + N_X} \sum_{j=1}^{N_X} \exp \frac{-||f_{\theta_s}(x_i) - f_{\theta_s}(x_j)||_2^2}{\tau} \\
& -\frac{1}{N_Y + N_X} \sum_{i=1}^{N_X} \frac{1}{N_Y + N_X} \sum_{j=1}^{N_Y} \exp \frac{-||f_{\theta_s}(x_i) - f_{\theta_s}(y_j)||_2^2}{\tau}
\end{aligned}
\tag{30}
$$

Importantly, the information-less hypothesis constrains the specific encoder to produce background embeddings aligned on the information-less vector $s'$. This property implies that background samples should not have any variability expressed in the latent space. Assuming that the salient encoder respects this property yields $f_{\theta_S}(t(x)) = s'$, it enables to express $\hat{H}(S)$ as:

$$
-\exp \mathcal{L}_{s'\text{uniform}} = 2 \frac{1}{N_Y + N_X} \sum_{i=1}^{N_Y} \frac{N_X}{N_Y + N_X} \exp \frac{-||f_{\theta_s}(y_i) - s'||_2^2}{\tau} + \frac{N_X}{N_Y + N_X} \frac{N_X}{N_Y + N_X}
$$
$$
\frac{1}{N_Y + N_X} \sum_{i=1}^{N_Y} \frac{1}{N_Y + N_X} \sum_{j=1}^{N_Y} \exp \frac{-||f_{\theta_s}(y_i) - f_{\theta_s}(y_j)||_2^2}{\tau}
$$
(31)

Assuming that the target and background datasets are balanced: $N_X = N_Y = N$ and ignoring the constant terms, we obtain:

$$
\mathcal{L}_{s'\text{uniform}} = -\log \frac{1}{N_Y} \sum_{i=1}^{N_Y} \left( \exp \frac{-||f_{\theta_s}(y_i) - s'||_2^2}{\tau} + \frac{1}{2N_Y} \sum_{j=1}^{N_Y} \exp \frac{-||f_{\theta_s}(y_i) - f_{\theta_s}(y_j)||_2^2}{\tau} \right)
$$
(32)

### B.3 ON THE INFORMATION-LESS HYPOTHESIS:

To respect the Information-less hypothesis, we re-write Eq. 5 as a Lagrangian function, with the constraint expressed as a $\beta$-weighted ($\beta \geq 0$) KL regularization. Assuming that $s_x$ follows a Gaussian distribution centered on $f_{\theta_s}(x)$ with a constant standard deviation $\tau$ permits deriving the KL divergence into an L2-distance between $f_{\theta_s}(x)$ and $s'$. Let us re-write Eq. 5 under the KKT conditions:

$$
-\mathcal{F}(\theta_S, \beta; x, y, s) = \mathcal{L}_{y\text{-alignment}} + \mathcal{L}_{s'\text{-uniformity}} + \beta \frac{1}{N_X} \sum_{i=1}^{N_X} ||f_{\theta_s}(x_i) - s'||_2^2
$$
(33)

## C  RETRIEVE THE SUPERVISED INFONCE LOSS

The Supervised counterpart of the InfoNCE loss has been introduced in Khosla et al. (2020). Compared to SimCLR, it consists of choosing positive pairs from the same class, while the negative pairs term remains unchanged. Let $X = \{(x_i)\}_{i=1}^N$ be a data-set of images $x_i$, $Y = \{(y_i)\}_{i=1}^N$ be their associated discrete or continuous labels $y_i$, and $Z = \{(z_i)\}_{i=1}^N$ the associated latent codes $z_i$. Let us introduce the maximization of Mutual Information between the labels $Y$ and latent vectors $Z$. The Mutual Information can be decomposed as follows:

$$
I(z; y) = \underbrace{-\mathbf{E}_{y \sim p_y} H(z|y)}_{\text{Supervised Alignment term}} + \underbrace{H(z)}_{\text{Uniformity term}}
$$
(34)

The Supervised counterpart of the InfoNCE loss has been introduced in Khosla et al. (2020). In this section, we show that it can be derived from the MI between $Y$ and $f_\theta(t(X))$. Compared to InfoNCE, it consists in aligning positive views $(t(x_i), t(x_j))$ from the same class $y_i = y_j$ via a supervised alignment term $-\mathbf{E}_{y \sim p_y} H(z|y)$, while the entropy term estimation $H(z)$ remains the same. Using the re-substitution estimator and the KDE, we derive the supervised alignment term into the alignment term of $\mathcal{L}_{\text{sup}}^{\text{in}}$ in Khosla et al. (2020):

$$
-\mathbf{E}_{y \sim p_y} H(z|y) + \log(\sqrt{2\pi\tau}) = \frac{1}{N} \sum_{i=1}^N \log \frac{1}{|P(i)|} \sum_{j \in P(i)} \exp \frac{-||f_\theta(v_i)^T - f_\theta(v_j)||_2^2}{2\tau}
$$
(35)

where $P(i)$ is the set of indices of samples belonging to class $y_i$ and $|P(i)|$ is its cardinality.
In Dufumier et al. (2021a), the authors proposed a generalized version of SupCon, which accounts for continuous label $y$.

## C.1 ON THE DISTINCTION BETWEEN $\mathcal{L}_{\text{SUP}}^{\text{IN}}$ AND $\mathcal{L}_{\text{SUP}}^{\text{OUT}}$:

In Khosla et al. (2020), the authors show that it is preferable to optimize $\mathcal{L}_{\text{sup}}^{\text{out}}$, a variant of $\mathcal{L}_{\text{sup}}^{\text{in}}$ where positive samples are summed outside of the logarithm. We propose to derive $\mathcal{L}_{\text{sup}}^{\text{out}}$ rather than $\mathcal{L}_{\text{sup}}^{\text{in}}$ by simply computing a lower bound of the Alignment term via Jensen's inequality:

$$-\mathbf{E}_{y \sim p_y} H(z|y) + \log(\sqrt{2\pi\tau}) \geq -\frac{1}{N} \sum_{i=1}^{N} \frac{1}{|P(i)|} \sum_{j \in P(i)} \frac{||f_\theta(v_i)^T - f_\theta(v_j)||_2^2}{2\tau} \tag{36}$$

## C.2 QUANTIFY THE JENSEN GAP FOR SUPINFONCE:

In Eq.36, we derived a lower bound of the Conditional Entropy of the Supervised InfoMax formulation via Jensen's inequality. In this paragraph, we propose to a) quantify Jensen's Gap between both formulations and b) describe under which condition these formulations are equal (tight bound). The Jensen's Gap can be computed as:

$$J_{\text{GAP}} = \frac{1}{N} \sum_{i=1}^{N} \log \underbrace{\frac{1}{|P(i)|} \sum_{j \in P(i)} \exp \frac{-||f_\theta(v_i)^T - f_\theta(v_j)||_2^2}{2\tau} - \frac{1}{|P(i)|} \sum_{j \in P(i)} \frac{-||f_\theta(v_i)^T - f_\theta(v_j)||_2^2}{2\tau}}_{\text{D}_{\text{GAP}}^{\text{sup}}} \tag{37}$$

where $J_{\text{GAP}} \geq 0$. Let us note $J_{\text{GAP}} = 0$ if and only if $\text{D}_{\text{GAP}}^{\text{sup}} = 0$. We simplified $\text{D}_{\text{GAP}}^{\text{sup}}$ into the difference between a LogSumExp and a SumLogExp of $f_\theta(v_i)^T . f_\theta(v_j')$. Using the fact that LogSumExp consists of a smooth approximation of the max function, $\text{D}_{\text{GAP}} = 0$ if and only if:

$$\max_j ||f_\theta(v_i)^T - f_\theta(v_j)||_2^2 + \log \frac{1}{N} = \frac{1}{N} \sum_{j=1}^{N} ||f_\theta(v_i)^T - f_\theta(v_j)||_2^2 \quad , \forall i \text{ in } |[1, N]| \tag{38}$$

where $y_i = y_j$, *i.e:* $v_i$, $v_j$ and $v_j'$ are views from images from the same class $y$.

## C.3 THE CASE OF A CONTINUOUS $y$:

In Dufumier et al. (2021a), the authors proposed a generalized version of SupCon, which accounts for continuous label $y$. It consists of adding a weight $w_\sigma(y_i, y_j)$ before the similarity term. Let us explain how to retrieve this formulation. From Eq. 34, we use the resubstitution estimator:

$$-\mathbf{E}_{y \sim p_y} H(z|y) + \log(\sqrt{2\pi\tau}) = \frac{1}{N} \sum_{i=1}^{N} \log \hat{p}(z_i|y_i) \tag{39}$$

From there, we can use a Kernel Density estimation for the conditional distributions in the case where we only have access to samples from the joint distribution:

$$-\mathbf{E}_{y \sim p_y} H(z|y) + \log(\sqrt{2\pi\tau}) = \frac{1}{N} \sum_{i=1}^{N} \log \frac{\frac{1}{N} \sum_{j=1}^{N} K_Y(y_i, y_j) K_Z(f_\theta(x_i), f_\theta(x_j))}{\frac{1}{N} \sum_{j=1}^{N} K_Y(y_i, y_j)} \tag{40}$$

By choosing $K_Y$ as a gaussian kernel: $K_y(y_i, y_j) = \frac{1}{\sigma\sqrt{2\pi}} \exp -\frac{(y_i - y_j)^2}{2\sigma^2}$ and $K_Z$ as a von Mises-Fisher kernel as usually done in Contrastive Learning literature, we retrieve Dufumier et al. (2021a)'s $L_{\text{sup}}^{\text{in}}$ formulation.

$$-\mathbf{E}_{y \sim p_y} H(z|y) + \log(\sqrt{2\pi\tau}) = \frac{1}{N} \sum_{i=1}^{N} \log \frac{1}{N} \sum_{j=1} w_\sigma(y_i, y_j) \exp \frac{-||f_\theta(x_i) - f_\theta(x_j)||_2^2}{2\tau} \tag{41}$$

where $w_\sigma(y_i, y_j) = \frac{K_Y(y_i, y_j)}{\frac{1}{N} \sum_{j=1} K_Y(y_i, y_j)}$.

Now, the Jensen's inequality can be utilized to retrieve Dufumier et al. (2021a)'s exact formulation.

## D   MAXIMIZE THE JOINT ENTROPY VIA KERNEL DENSITY-BASED ESTIMATION

In Sec. 3.3, we proposed a method to estimate and minimize $-H(c,s)$ without requiring any assumptions about the form of its pdf nor requiring a neural network-based approximation (Cheng et al. (2020); Alemi (2020); Poole et al. (2019)). Inspired by Holmes et al. (2007), we develop $H(c,s)$ with a re-substitution entropy estimator:

$$-\hat{H}(c,s) = \frac{1}{N_X + N_Y} \sum_{i=1}^{N_X+N_Y} \log \hat{p}(c_i, s_i) \quad (42)$$

To do so, we estimate the density $\hat{p}_\theta(c_i, s_i)$ with a Kernel Density Estimation:

$$-\hat{H}(c,s) = \frac{1}{N_X + N_Y} \sum_{i=1}^{N_X+N_Y} \log \frac{1}{N_X + N_Y} \sum_{k=1}^{N_X+N_Y} K_C(c_i, c_j) K_S(s_i, s_j) \quad (43)$$

where $c_j$ and $s_j$ are drawn from the joint distribution $p(c,s)$. In practice, we will draw pairs $(c,s)$ from $(f_{\theta_C}(x), f_{\theta_S}(x))$ and $(f_{\theta_C}(y), f_{\theta_S}(y))$ where $x$ and $y$ are respectively uniformly drawn from $X$ and $Y$. Importantly, as in Sec. 3.2, the information-less constraint still holds: $f_{\theta_S}(x) = s', \forall x$. For simplicity, we choose Gaussian kernels for $K_C$ and $K_S$ with a constant standard deviation parameter $\tau$, which simplifies the estimation of the joint entropy into:

$$-\hat{H}(c,s) = \frac{1}{N_X + N_Y} \sum_{i=1}^{N_X+N_Y} \log \frac{1}{N_X + N_Y} \sum_{j=1}^{N_X+N_Y} \exp \frac{-||c_i - c_j||_2^2}{2\tau} \exp \frac{-||s_i - s_j||_2^2}{2\tau} \quad (44)$$

## E   CAPTURING INDEPENDENT ATTRIBUTES AND DISENTANGLE WITH CONTRASTIVE LEARNING

### E.1   SUPERVISED DISENTANGLEMENT

We can also use our framework to derive a supervised disentangling loss with known variability factors. In this section, we propose to explore an extension of BC-InfoNCE in the case where independent fine-grained attributes about the target dataset: $\{a_i \in \mathcal{R}^{D_S}\}_{i=1}^{N_Y}$ are available. Given this set of independent observed characteristics, we can leverage these observations in a supervised manner to identify the independent factors of generation of the target dataset.

We assume the existence of $D_S$ attributes and construct our salient encoder so that it outputs $D_S$ latent dimensions. Our objective is to construct a salient space where each salient latent dimension $S^{d_s}$ only depends on its corresponding attribute $a^{d_s}$. Let us re-write Eq. 1 by replacing the salient InfoMax term by each d-th attribute Supervised InfoMax term:

$$\arg\max I(x;c) + I(y;c) + \frac{1}{D_S} \sum_{d=1}^{D_s} \underbrace{I(a^d; s^d)}_{\text{d-th SupInfoMax}} \quad \text{s.t. } D_{KL}(s_x || \delta(s')) = 0 \text{ and } I(c,s) = 0 \quad (45)$$

From there, we take inspiration from Dufumier et al. (2021a) to decompose each d-th attribute Supervised InfoMax term in a supervised alignment and a uniformity term:

$$I(a^d; s^d) \geq \frac{1}{N_Y} \sum_{i=1}^{N_Y} w_\sigma(a_i^d, a_j^d) \frac{||s_i^d - s_j^d||_2}{2\tau} + \hat{H}(s^d) = \mathcal{L}_{\text{d-th SupInfoMax}} \quad (46)$$

We propose to develop the Entropy term for each $d$-th salient dimension as in Sec. C.3.

## F   DATASETS AND IMPLEMENTATION DETAILS

### F.1   DSPRITES WATERMARKED ON A GRID OF DIGITS EXPERIMENT

To evaluate the Contrastive Analysis method enriched with target attributes, we provide a novel toy dataset. The background dataset $X$ consists of 4 MNIST digits (1-4) regularly placed on a grid.

The target dataset $Y$ consists of a dSprites item added upon the foreground of this grid of digits. dSprites is a dataset introduced to evaluate disentanglement. Its images are of size 64x64 pixels. Its elements only exhibit 5 generation factors, see Fig. 3, making it easy to evaluate the disentanglement. Possible variations are 1) shape (heart, ellipse, and square), 2) size, 3) position in X, 4) position in Y, and 5) orientation (i.e. rotation). To construct the Contrastive Analysis dataset we use in this paper, we randomly sample MNIST images of digits 1, 2, 3, and 4 and regularly place them on a grid. We create 25,000 background images with this method. Then, we superimpose a random dSprite element on 25,000 distinct digit grids to create 25 000 target images. We use the same method to derive 5,000 test images equally balanced between the target and background classes. Importantly, we constrain the dSprites elements to have a rotation attribute between $-45$ and $+45$ degrees. Downstream task performances are computed on the projection head.

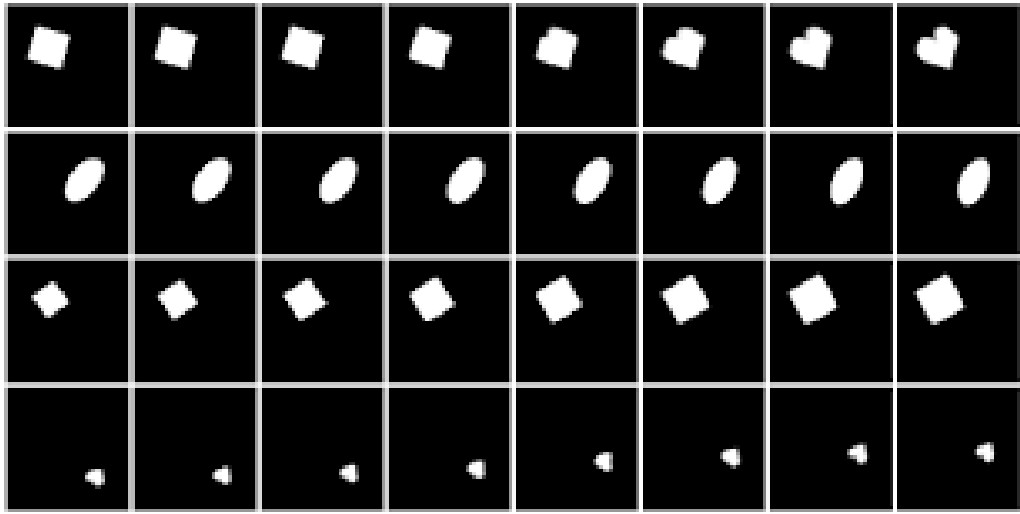

Figure 3: Illustration of the dSprites dataset and its different independent variability factors: shape, zoom, rotation, Y position, and X position.

## F.2 MNIST DIGIT SUPERIMPOSED ON CIFAR-10 BACKGROUND

MNIST digit superimposed on CIFAR-10 background is a simple intuitive dataset inspired from Zou et al. (2013). We consider as the background dataset ($y = 0$) CIFAR-10 images, and as the target dataset ($y = 1$) CIFAR-10 images (background) with an overlaid digit (target pattern), see Fig. 4. This experiment is particularly suited to CA, we expect our model to successfully capture the background variability (*i.e:* natural objects semantic) in the common space and to capture the digits variability in the salient space. In practice, we used a train set of 50000 images (25000 Cifar-10 images, 25000 Cifar-10 images with random MNIST digits overlaid) and an independent test set of 1000 images (500, 500). Images are of size $32 \times 32$. Pixels were normalized between 0 and 1.

In terms of Data Augmentation for the stochastic transformation process $t(.)$, we remained close to SimCLR Chen et al. (2020), as we used a RandomCrop(size=(24, 24), scale=(0.2, 1.0)) augmentation, then a RandomHorizontalFlip(p=0.5) augmentation, a RandomColorJitter(0.4, 0.4, 0.4, 0.1) applied with a probability 0.8 followed from a RandomGrayScale(p=0.2) augmentation.

Concerning the Neural Network architecture, both common and salient encoders were chosen as ResNet-18 with a representation linear layer as follows: linear(512, 32) and a non-linear projector layer as follows: (linear(32, 128), batch norm(128), relu(), linear(128, 32)). We used an Adam optimizer with learning rate of 5e-4, batch size of 512, and trained it during 250 epochs.

As for the SepCLR's hyper-parameters, we chose $\lambda_C = 1$, $\lambda_S = \beta = 1000$, and $\lambda = 10$. As related works, downstream task performances are computed before the projection head Chen et al. (2020).

Concerning Contrastive Analysis VAE methods, we took inspiration from experimental setups in Louiset et al. (2023). Namely, we used a standard encoder architecture composed of 4 convolutions (channels 3, 32, 32, 32, 256), kernel size 4, and padding (1, 1, 1, 0). Then, for each mean and

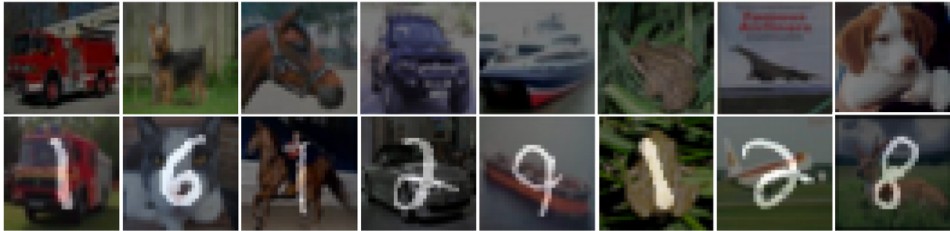

Figure 4: the Superimposed MNIST digits on CIFAR background dataset. Target images are CIFAR-10 images overlaid with an MNIST digit. Background images are CIFAR-10 images.

standard deviations predicted (common and salient), we used two linear layers going from 256 to hidden size 256 to (common and salient) latent space size 32. The decoder was set symmetrically. We used the same architecture across all the CA-VAEs concurrent works we evaluated. Interestingly, we also tried with ResNet-18 encoders but the results actually remained similar. The learning rate was set to 0.001 with an Adam optimizer. The models were trained during 250 epochs with batch size equal to 512. We used $\beta_c = 0.5$ and $\beta_s = 0.5$, $\kappa = 2$, $\gamma = 1e - 9$, $\alpha = \frac{1}{0.025}$. For cVAE, we used $\beta_c = 0.5$ and $\beta_s = 0.5$, $\kappa = 2$ and $\kappa = 0$ for conVAE. For MM-cVAE we used the same learning rate, $\beta_c = 0.5$ and $\beta_s = 0.5$, the background salient regularization weight 100, common regularization weight of 1000.

Concerning Mutual Information minimization methods, we used the same hyper-parameters as for k-JEM, except for $\lambda$. $\lambda$ was set to 0.1 for CLUB, as in the original paper Domain Adaptation section Cheng et al. (2020). Please note that we also tried values of 1 and 10, but it did not give better results. We also chose 0.1 for vUB and vL1out. For TC, we used $\lambda = 10$. For MMD, we used $\lambda = 50$; we motivate this choice in the Sec. G.1.

### F.3   CELEBA ACCESSORIES

In CelebA with accessories Weinberger et al. (2022), we consider a subset of CelebA Liu et al.. It contains two sets, target and background, from a subset of CelebA Liu et al., one with images of celebrities wearing glasses or hats (target) and the other with images of celebrities not wearing any of these accessories (background). Importantly, and contrarily to MM-cVAE Weinberger et al. (2022) and SepVAE Louiset et al. (2023), we take care to balance the distribution of males and females in the background and the target dataset to avoid gender bias with respect to the accesories. We used a train set of 20000 images, (10000 no accessories, 5000 glasses, 5000 hats) and an independent test set of 4000 images (2000 no accessories, 1000 glasses, 1000 hats). Images are of size $128 \times 128$, normalized between 0 and 1.

In terms of Data Augmentation for the stochastic transformation process $t(.)$, we remained close to SimCLR Chen et al. (2020), as we used a RandomCrop(size=(128, 128), scale=(0.2, 1.0)) augmentation, then a RandomHorizontalFlip(p=0.5) augmentation, a RandomColorJitter(0.4, 0.4, 0.4, 0.1) applied wit a probability 0.8 followed from a RandomGrayScale(p=0.2) augmentation.

Concerning the Neural Network architecture, both common and salient encoders were chosen as ResNet-18 with a representation linear layer as follows: linear(512, 16) and a non-linear projector layer as follows: (linear(16, 128), batch norm(128), relu(), linear(128, 16)). We used an Adam optimizer with learning rate 5e-4, batch size of 256, and trained it during 250 epochs.

As for the SepCLR's hyper-parameters, we chose, as in MNIST superimposed on CIFAR-10 experiment, $\lambda_C = 1$, $\lambda_S = \beta = 1000$, and $\lambda = 10$. As related works, downstream task performances are computed before the projection head Chen et al. (2020).

Concerning Contrastive Analysis VAE methods, we took inspiration from experimental setups in Louiset et al. (2023). Notably, we used images of size $64x64$ pixels. Namely, we use a standard encoder architecture composed of 5 convolutions (channels 3, 32, 32, 64, 128, 256), kernel size 4, stride 2, and padding (1, 1, 1, 1, 1). Then, concerning the mean and standard deviations predicted (common and salient), we used two linear layers going from 256 to hidden size 32 to (common and salient) latent space size 16. The decoder was set symmetrically. We used the same architecture across all the CA-VAEs concurrent works we evaluated. The learning rate was set to 0.001 with an Adam optimizer. The models were trained during 250 epochs with batch size equal to 512. We used

$\beta_c = 0.5$ and $\beta_s = 0.5$, $\kappa = 2$, $\gamma = 1e - 10$, $\sigma_p = 0.025$. For cVAE, we used $\beta_c = 0.5$ and $\beta_s = 0.5$, $\kappa = 2$ and $\kappa = 0$ for conVAE. For MM-cVAE, we used the same learning rate, $\beta_c = 0.5$ and $\beta_s = 0.5$, the background salient regularization weight 100, common regularization weight of 1000.

Concerning Mutual Information minimization methods, we used the same hyper-parameters as for k-JEM, except for $\lambda$. $\lambda$ was set to 0.1 for CLUB, as in the original paper Domain Adaptation section Cheng et al. (2020). Please note that we also tried values of 1 and 10, but it did not give better results. We also chose 0.1 for vUB and vL1out. For TC, we used $\lambda = 10$. For MMD, we used $\lambda = 50$; we motivate this choice in the Sec. G.1.

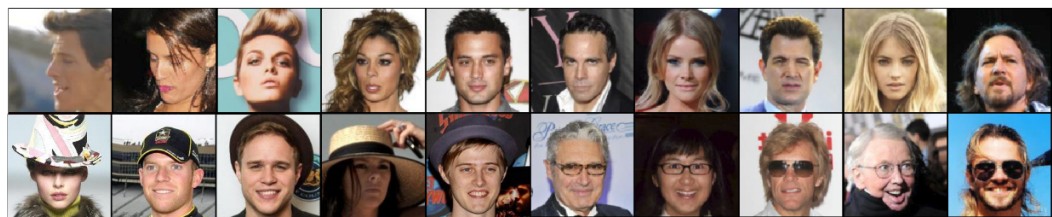

Figure 5: Celeba accessories dataset. The upper row consists of background images. The lower row shows target images.

## F.4 CHEXPERT

In the CheXpert subtyping experiment, we select a subset of CheXpert separated in the background dataset: 10,000 healthy X-rays and the target dataset: 3,000 with edema, 3,000 with pleural effusion, and around 2,000 images with cardiomegaly. Images are resized to 224x224 pixels. Pixels are normalized between 0 and 1.

For SepCLR, in terms of Data Augmentation for the stochastic transformation process $t(.)$, we remained close to SimCLR Chen et al. (2020), as we used a RandomCrop(size=(224, 224), scale=(0.2, 1.0)) augmentation, a RandomColorJitter(0.4, 0.4, 0.4, 0.1) applied with a probability 0.8 followed from a RandomGrayScale(p=0.2) augmentation, a RandomRotation(degrees=45), and then a RandomHorizontalFlip(p=0.5) augmentation.

Concerning the Neural Network architecture, both common and salient sizes are 32. Both common and salient encoders were chosen as a pre-trained ResNet-18 with a representation linear layer as follows: linear(512, 32) and a non-linear projector layer as follows: (linear(32, 128), batch norm(128), relu(), linear(128, 32)). We used an Adam optimizer with a learning rate of 5e-4, a batch size of 256, and trained it during 200 epochs. As for the SepCLR's hyper-parameters, we chose $\lambda_C = 1$, $\lambda_S = 1$, $\beta = 10$, and $\lambda = 5$. As related works, downstream task performances are computed before the projection head Chen et al. (2020).

Concerning the Contrastive VAEs, we use the same common and salient encoders. For the decoders, we chose an architecture composed of a linear layer, taking into input the concatenation of common and salient space, mapping it to a size of 256. Then 7 deconvolution layers were used with a kernel size of 4, stride of 2, and padding of 1 with filters (256 to 512, 256, 128, 64, 32, 16, 3). Output images are of size $256 \times 256$ and are cropped to $224 \times 224$. The final activation layer is chosen as a sigmoid layer.

We used the same architecture across all the CA-VAEs concurrent works we evaluated. The learning rate was set to 0.001 with an Adam optimizer. The models were trained during 200 epochs with batch size equal to 256. We used $\beta_c = 0.5$ and $\beta_s = 0.5$, $\kappa = 2$, $\gamma = 1e - 9$, $\sigma_p = 0.05$. For cVAE, we used $\beta_c = 0.5$ and $\beta_s = 0.5$, $\kappa = 2$ and $\kappa = 0$ for conVAE. For MM-cVAE, we used the same learning rate, $\beta_c = 0.5$ and $\beta_s = 0.5$, the background salient regularization weight 100, common regularization weight of 1000.

## F.5 ODIR (OCULAR DISEASE IMAGE RECOGNITION)

In the ODIR subtyping experiment, we select a subset of the ODIR dataset separated into a background and a target dataset. Train dataset contains 1890 healthy images, 363 diabetes images,

278 glaucoma images, 281 cataract images, 242 age-related macular degeneration images, and 227 pathological myopia images. On the other hand, TEST dataset contains respectively 210 healthy, 37 diabetes, 26 glaucoma, 39 cataract, 23 macular degeneration, 30 myopia images. Pixels are normalized between 0 and 1.

For SepCLR, in terms of Data Augmentation for the stochastic transformation process $t(.)$, we remained close to SimCLR Chen et al. (2020), as we used a RandomCrop(size=(224, 224), scale=(0.75, 1.0)) augmentation, a RandomColorJitter(0.4, 0.4, 0.4, 0.1) applied wit a probability 0.8 followed from a RandomGrayScale(p=0.2) augmentation, a RandomRotation(degrees=45), and then a RandomVerticalFlip(p=0.5) augmentation.

Concerning the Neural Network architecture, both common and salient encoders were chosen as a pre-trained ResNet-18 with a representation linear layer as follows: linear(512, 32) and a non-linear projector layer as follows: (linear(32, 128), batch norm(128), relu(), linear(128, 32)). We used an Adam optimizer with a learning rate of 5e-4, a batch size of 256, and trained it during 200 epochs. As for the SepCLR's hyper-parameters, we chose $\lambda_C = 1$, $\lambda_S = 1$, $\beta = 100$, and $\lambda = 10$. As related works, downstream task performances are computed before the projection head Chen et al. (2020).

Concerning the Contrastive VAEs, we use the same common and salient encoders. For the decoders, we chose an architecture composed of a linear layer, taking into input the concatenation of common and salient space, mapping it to a size of 256. Then 7 deconvolution layers were used with a kernel size of 4, stride of 2, and padding of 1 with filters (256 to 512, 256, 128, 64, 32, 16, 3). Output images are of size $256 \times 256$ and are cropped to $224 \times 224$. The final activation layer is chosen as a sigmoid layer.

We used the same architecture across all the CA-VAEs concurrent works we evaluated. The learning rate was set to 0.001 with an Adam optimizer. The models were trained during 200 epochs with batch size equal to 256. We used $\beta_c = 0.5$ and $\beta_s = 0.5$, $\kappa = 2$, $\gamma = 1e - 9$, $\sigma_p = 0.05$. For cVAE, we used $\beta_c = 0.5$ and $\beta_s = 0.5$, $\kappa = 2$ and $\kappa = 0$ for conVAE. For MM-cVAE, we used the same learning rate, $\beta_c = 0.5$ and $\beta_s = 0.5$, the background salient regularization weight 100, common regularization weight of 1000.

### F.6 SCHIZOPHRENIA EXPERIMENT

In this study, we analyzed neuroimaging data from several sources including the SCHIZCONNECT database (which includes 368 healthy controls and 275 patients with schizophrenia) and the BSNIP database (which includes 199 healthy controls and 190 patients with schizophrenia). The data used in this study was collected from various scanners and locations and included brain scans from individuals in the United States. Images are of size $128 \times 128 \times 128$ with voxels normalized on a Gaussian distribution per image. Experiments were run 5 times with a different train/val split (respectively 75% and 25% of the dataset) to account for initialization and data uncertainty. Inspired by Louiset et al. (2023), common and salient convolutional encoders were chosen as 5 3D-convolutions (channels 1, 32, 64, 128, 256, 512), kernel size 4, stride 2, and padding 1 followed by batch normalization layers. Then, we used a linear layer from 32768 to representations (sizes 128 for common and 32 for salient). Then, the projection heads were set as non-linear with hidden sizes 128 for common and 32 for salient, with batch normalization(128) and relu() activation functions.

For SepCLR, the data augmentations were inspired from Dufumier et al. (2021a), that is: horizontal flip with probability 0.5; blur with probability 0.5, sigma=(0.1, 0.1); noise with probability 0.5, sigma=(0.1, 0.1); CutOut with probability 0.5, patch size equal to 32x32x32, RandomCrop of size (96x96x96) with probability 0.5. The models were trained during 50 epochs with a batch size equal to 32 with an Adam optimizer of learning rate of 0.0005. As for the SepCLR's hyper-parameters, we chose $\lambda_C = 1$, $\lambda_S = 1$, $\beta = 1$, and $\lambda = 5$. As related works, downstream task performances are computed before the projection head Chen et al. (2020). Importantly, the classification task is computed with a 2 layers MLPs in order to be comparable with SepVAE Louiset et al. (2023)

Concerning the Contrastive Analysis VAEs methods we compared with, we use the same experimental setup in terms of hyper-parameters and architecture as in Louiset et al. (2023). Concerning the architecture, in details, the common and salient convolutional encoders were chosen as 5 3D-convolutions (channels 1, 32, 64, 128, 256, 512), kernel size 4, stride 2, and padding 1 followed by batch normalization layers. Then, we used a non linear layer from 32768 to directly predict mean and standard deviations (sizes 256 for common and 256 for salient) with 2048 as hidden

size with batch normalization and relu as activation functions. The decoder was set symmetrically, except it has 6 transposed convolutions (channels 512, 256, 128, 64, 32, 16, 1), kernel size 3, stride 2, and padding 1, followed by batch normalization layers.

### F.7 MUTUAL INFORMATION MINIMIZATION METHODS

To compare with k-JEM (kernel-Joint Entropy Maximization), we used the implementation of several Mutual Information variational upper bound, namely vCLUB Cheng et al. (2020), vUB Alemi (2020) and vL1out Poole et al. (2019) available at `https://github.com/Linear95/CLUB/tree/master`. Interestingly, these methods can be implemented with a variational approximation of $S$ from $C$, vice-versa ($C$ from $S$), or symmetrically (mean of both). We tried all three possibilities with different weights and chose the best results each time to set in Tab .1 and Tab .2.
We also compared with the exact Mutual Information estimator TC of Louiset et al. (2023) and Abid & Zou (2019) inspired by the Total Correlation introduced in Kim & Mnih (2018).
In Sec. 3.3, we motivated the idea of minimizing the negative joint entropy ($-H(C, S)$) rather than the Mutual Information ($H(C) + H(S) - H(C, S)$). To prove our point, we implemented k-MI, a version of k-JEM where we also minimize the entropies $H(C) + H(S)$. To do so, we estimate $H(C)$ as in the common entropy estimation in Eq. 15 and $H(S)$ as in the salient entropy estimation in Eq. 29. Interestingly, we can see that k-MI indeed underperforms compared to k-JEM.

## G SUPPLEMENTARY RESULTS

### G.1 ON MUTUAL INFORMATION MINIMIZATION VERSUS TARGET AND BACKGROUND DISTRIBUTIONS MATCHING

In Contrastive Analysis, practitioners make use of various regularizations to respect properties established a priori. Recent works agree that background input should be mapped to a single information-less vector in the salient space. However, two regularizations have been proposed in order to reduce the information leakage between the common and the salient space: 1- match the distributions of targets and backgrounds in the common space, 2- minimize the mutual information between the common and salient distributions. In our framework, the latter was naturally derived from the Info-Max principle. In Tab .6 and Tab .7, we propose to compare both strategies on a) CelebA accessories and b) Digits superimposed on CIFAR-10 to assess their effect on the common space. In both experiments, we observe that the stronger the regularization is, the less common information (objects and sex) is captured. Also, we observe that k-JEM 's ability to diminish target-specific information (digits and accessories) remains relatively consistent across the regularization strength. Concerning MMD (Maximum Mean Discrepancy), a high regularization strength is needed to reduce target-specific information despite its detrimental effect on capturing common patterns. We conclude that a low-strength k-JEM regularization (we choose $\lambda = 10$ in practice) is the right trade-off for capturing common patterns while canceling salient patterns.

Table 6: Digits watermarked on CIFAR-10 (B-ACC). Comparison of k-JEM with MMD given different strengths.

|  | DIGITS | | OBJECTS | | $\delta_{\text{TOT}} \downarrow$ |
|---|---|---|---|---|---|
|  | S $\uparrow$ | C $\downarrow$ | S $\downarrow$ | C $\uparrow$ | |
| SEPCLR-NO K-JEM | 95.6 | 94.4 | 9.0 | 42.0 | 145.8 |
| SEPCLR-10 MMD | 95.4 | 86.8 | 10.8 | 48.2 | 56.8 |
| SEPCLR-50 MMD | 94.6 | 21.2 | 9.0 | 62.2 | 134.0 |
| SEPCLR-100 MMD | 95.2 | 13.8 | 11.0 | 56.4 | 53.2 |
| SEPCLR-10 K-JEM | 96.2 | 11.0 | 10.4 | 73.2 | **32.0** |
| SEPCLR-50 K-JEM | 95.2 | 13.2 | 8.6 | 59.2 | 47.4 |
| SEPCLR-100 K-JEM | 95.0 | 12.0 | 9.2 | 52.4 | 53.8 |
| BEST EXPECTED | 100.0 | 10.0 | 10.0 | 100.0 | 0.0 |

Table 7: CelebA accessories (B-ACC). Comparison of k-JEM with MMD given different strengths.

| | HATS/GLSS | | SEX | | $\delta_{\text{TOT}} \downarrow$ |
|---|---|---|---|---|---|
| | S $\uparrow$ | C $\downarrow$ | S $\downarrow$ | C $\uparrow$ | |
| SEPCLR-NO K-JEM | 99.03 | 66.68 | 98.48 | 79.48 | 86.65 |
| SEPCLR-10 MMD | 98.99 | 81.53 | 60.87 | 76.19 | 87.14 |
| SEPCLR-50 MMD | 98.95 | 67.50 | 65.47 | 71.83 | 62.19 |
| SEPCLR-100 MMD | 99.03 | 53.25 | 67.12 | 52.51 | 68.83 |
| SEPCLR-10 K-JEM | 98.57 | 55.21 | 62.52 | 78.00 | **41.16** |
| SEPCLR-50 K-JEM | 98.83 | 58.27 | 62.45 | 68.38 | 53.51 |
| SEPCLR-100 K-JEM | 98.73 | 62.00 | 68.92 | 57.10 | 75.09 |
| BEST EXPECTED | 100.0 | 50.0 | 50.0 | 100.0 | 0.0 |

## G.2 ON THE ADD OF A RECONSTRUCTION TERM

Contrastive Analysis, jointly performed with a generative process, enables performing salient or common characteristics swapping, salient attribute generation or deletion, and novel sample generation. Therefore, we investigated the addition of a decoder jointly trained with the encoder parameters to reconstruct the input images (with a Mean Square Error Cost Function) during the optimization process. We added a reconstruction term from the concatenation of the common and salient space (as in CA-VAEs but without the need for a re-parameterization trick) with the same decoder as in CA-VAEs, and it degrades the results. Intriguingly, we found that it tends to degrade the results (see Tab. 9 and Tab. 8), which could be explained by the fact that the reconstruction task tries to conserve unnecessary noisy information in the latent space. However, an interesting perspective could be to include and train a generator or a decoder for generation and interpretability purposes, given frozen representations learned with SepCLR.

Table 8: Digits watermarked on CIFAR-10 (B-ACC). On the impact of a reconstruction term in addition to SepCLR.

| | DIGITS | | OBJECTS | | $\delta_{\text{TOT}} \downarrow$ |
|---|---|---|---|---|---|
| | S $\uparrow$ | C $\downarrow$ | S $\downarrow$ | C $\uparrow$ | |
| SEPCLR-K-JEM | 96.2 | 11.0 | 10.4 | 73.2 | **32.0** |
| SEPCLR-K-JEM WITH 0.1 REC | 98.8 | 10.8 | 40.6 | 47.2 | 85.4 |
| SEPCLR-K-JEM WITH 1 REC | 94.4 | 22.2 | 51.8 | 27.4 | 132.2 |
| BEST EXPECTED | 100.0 | 10.0 | 10.0 | 100.0 | 0.0 |

Table 9: CelebA accessories (B-ACC). On the impact of a reconstruction term in addition to SepCLR.

| | HATS/GLSS | | SEX | | $\delta_{\text{TOT}} \downarrow$ |
|---|---|---|---|---|---|
| | S $\uparrow$ | C $\downarrow$ | S $\downarrow$ | C $\uparrow$ | |
| SEPCLR-K-JEM | 98.57 | 55.21 | 62.52 | 78.00 | **41.16** |
| SEPCLR-K-JEM WITH 0.1 REC | 97.27 | 67.81 | 67.53 | 62.38 | 75.69 |
| SEPCLR-K-JEM WITH 1 REC | 91.51 | 68.87 | 62.77 | 64.39 | 78.98 |
| BEST EXPECTED | 100.0 | 50.0 | 50.0 | 100.0 | 0.0 |

## G.3 ON THE COMPARISON WITH CONTRASTIVE METHODS

In this section, we propose to compare SepCLR with self-supervised methods that are not based on the encoder-decoder architecture. As there are no Contrastive Learning methods tailored for Contrastive Analysis, we propose to design a naive and simple strategy to compare with SepCLR. First,

we infer the common features with the features of SimCLR trained on the background dataset only (as it should have common features only). Then, we propose to infer the salient space with a SupCon method trained to discriminate the background samples from the target samples. This way, such a method should capture target-specific patterns while discarding common features. Additionally, we compare with SimCLR trained on both datasets to get a reference point (even though, in that case, the common space and the salient space are the same unique space, which cannot perform the separation of common and salient patterns). See Tab. 10, Tab. 11 and Tab. 12 for the results. SepCLR always performs better in terms of $\delta_{\text{tot}}$.

Table 10: Comparison of SepCLR-k-JEM with Contrastive methods on Digits watermarked on CIFAR-1 (B-ACC)

|  | DIGITS | | OBJECTS | | $\delta_{\text{TOT}} \downarrow$ |
| --- | --- | --- | --- | --- | --- |
|  | S $\uparrow$ | C $\downarrow$ | S $\downarrow$ | C $\uparrow$ | |
| SIMCLR ON TG AND BG | 44.0 | 44.0 | 94.6 | 94.6 | 180.0 |
| SIMCLR + SUPCON | 41.4 | 51.4 | 19.0 | 50.0 | 159.0 |
| SEPCLR-K-JEM | 96.2 | 11.0 | 10.4 | 73.2 | **32.0** |
| BEST EXPECTED | 100.0 | 10.0 | 10.0 | 100.0 | 0.0 |

Table 11: Comparison of SepCLR-k-JEM with Contrastive methods on CelebA accessories (B-ACC).

|  | HATS/GLSS | | SEX | | $\delta_{\text{TOT}} \downarrow$ |
| --- | --- | --- | --- | --- | --- |
|  | S $\uparrow$ | C $\downarrow$ | S $\downarrow$ | C $\uparrow$ | |
| SIMCLR ON BG AND TG | 98.92 | 98.92 | 84.16 | 84.16 | 100.0 |
| SIMCLR + SUPCON | 97.93 | 82.15 | 59.98 | 80.76 | 63.44 |
| SEPCLR-K-JEM | 98.57 | 55.21 | 62.52 | 78.00 | **41.16** |
| BEST EXPECTED | 100.0 | 50.0 | 50.0 | 100.0 | 0.0 |

Table 12: Comparison of SepCLR-k-JEM with Contrastive methods on ODIR dataset (B-ACC).

|  | SUBTYPE | | SEX | | $\delta_{\text{TOT}} \downarrow$ |
| --- | --- | --- | --- | --- | --- |
|  | S $\uparrow$ | C $\downarrow$ | S $\downarrow$ | C $\uparrow$ | |
| SIMCLR ON BG AND TG | 66.10 | 66.10 | 57.20 | 57.20 | 125.0 |
| SIMCLR + SUPCON | 68.70 | 57.17 | 51.94 | 58.41 | 107.0 |
| SEPCLR-K-JEM | 68.54 | 47.71 | 52.48 | 59.62 | **97.03** |
| BEST EXPECTED | 100.0 | 25.0 | 50.0 | 100.0 | 0.0 |

## G.4 ABLATION STUDY

In the main text, we investigated the effect of a null Mutual Information constraint by removing the proposed loss (No k-JEM) or by minimizing the Mutual Information estimate (k-MI) rather than the Joint Entropy estimate (see Tab. 1 and Tab. 2). Here, we propose a further ablation study in Tab. 13 and Tab. 14. We report the results of our method when removing all proposed losses one by one. We can observe that each loss is important since, when removing it, we either degrade the capture of salient patterns or we fail to disregard the common features in the salient space.

Table 13: Ablation Study on Digits watermarked on CIFAR-10 (B-ACC).

|  | DIGITS | | OBJECTS | | $\delta_{\text{TOT}} \downarrow$ |
| --- | --- | --- | --- | --- | --- |
|  | S $\uparrow$ | C $\downarrow$ | S $\downarrow$ | C $\uparrow$ | |
| SEPCLR-$\lambda = 0$ (NO K-JEM) | 95.6 | 94.4 | 9.0 | 42.0 | 145.8 |
| SEPCLR-$\beta = 0$ (NO INFOLESS REG) | 96.2 | 11.6 | 10.4 | 71.8 | 34.0 |
| SEPCLR-$\lambda_S = 0$ (NO SALIENT TERM) | 93.4 | 42.0 | 18.6 | 40.0 | 90.25 |
| SEPCLR-$\lambda_C = 0$ (NO COMMON TERM) | 94.4 | 10.4 | 18.8 | 20.4 | 94.4 |
| SEPCLR-K-JEM | 96.2 | 11.0 | 10.4 | 73.2 | **32.0** |
| BEST EXPECTED | 100.0 | 10.0 | 10.0 | 100.0 | 0.0 |

Table 14: Ablation Study on CelebA accessories (B-ACC).

| | Hats/Glss | | Sex | | $\delta_{\text{TOT}}\downarrow$ |
|---|---|---|---|---|---|
| | S $\uparrow$ | C $\downarrow$ | S $\downarrow$ | C $\uparrow$ | |
| SepCLR - $\lambda = 0$ (no k-JEM) | 99.03 | 66.68 | 98.48 | 79.48 | 86.65 |
| SepCLR - $\beta = 0$ (no Infoless reg) | 99.12 | 53.88 | 68.82 | 77.29 | 46.29 |
| SepCLR - $\lambda_S = 0$ (no Salient term) | 77.50 | 87.73 | 53.30 | 77.55 | 85.98 |
| SepCLR - $\lambda_C = 0$ (no Common term) | 98.38 | 56.32 | 66.44 | 53.09 | 71.29 |
| SepCLR - k-JEM | 98.57 | 55.21 | 62.52 | 78.00 | **41.16** |
| Best Expected | 100.0 | 50.0 | 50.0 | 100.0 | 0.0 |

## G.5 Performances on the background datasets

In the main text, we evaluated our method on the ability to linearly predict common attributes in the common space only and salient attributes in the salient space only on target samples only (as they are generated from both common and target-specific factors of variability). In this section, we evaluate the ability to linearly predict common attributes only in the common space. In Tab. 15 and Tab 16, we can see that the common performances remain good on background samples while the salient space is non-informative as it is supposed to be. We can also notice that, compared to concurrent CA-VAE methods, our method is still the best-performing one in terms of $\delta$, as it predicts common attributes way better.

Table 15: Balanced Accuracy results on Digits watermarked on CIFAR-10 on background samples.

| | Digits | | Objects | | $\delta\downarrow$ |
|---|---|---|---|---|---|
| | S $\uparrow$ | C $\downarrow$ | S $\downarrow$ | C $\uparrow$ | |
| MM-cVAE | $\times$ | $\times$ | 18.2 | 32.8 | 75.4 |
| SepVAE | $\times$ | $\times$ | 20.0 | 34.4 | 75.6 |
| SepCLR-k-JEM | $\times$ | $\times$ | 28.0 | 74.0 | **44.0** |
| Best expected | $\times$ | $\times$ | 10.0 | 100.0 | 0.0 |

Table 16: Balanced Accuracy results on CelebA accessories on background samples.

| | Hats/Glss | | Sex | | $\delta\downarrow$ |
|---|---|---|---|---|---|
| | S $\uparrow$ | C $\downarrow$ | S $\downarrow$ | C $\uparrow$ | |
| MM-cVAE | $\times$ | $\times$ | 64.27 | 70.48 | 43.79 |
| SepVAE | $\times$ | $\times$ | 56.42 | 70.19 | 36.23 |
| SepCLR - k-JEM | $\times$ | $\times$ | 64.10 | 86.63 | **27.47** |
| Best Expected | $\times$ | $\times$ | 50.0 | 100.0 | 0.0 |

## G.6 On the impact of $\mathcal{L}_{\text{UNIF}}$ or $\mathcal{L}_{\text{LOG-SUM-EXP}}$

As shown in Sec. A, we estimate the entropy using a resubstitution entropy estimator. This results in one of the terms of the standard Contrastive loss (i.e., InfoNCE) that accounts for the negative samples. As shown in Wang et Isola, this term has the same minimizer as the $L_{\text{unif}}$ loss when the number of negatives tends to be infinite. We decided to use the $L_{\text{unif}}$ loss instead of the Contrastive loss because it is computationally less expensive, and it has been shown by Wang et Isola to lead to good representations and good downstream task performance. Furthermore, we have also compared the two losses in the CIFAR10-MNIST dataset the CelebA accessories (see Tab. 17 and Tab. 18) and found that the results are slightly better or similar using $L_{\text{unif}}$.

## G.7 On the impact of the encoders

In this section, we justify our choices in terms of architecture. In Tab. 19 and Tab. 20, we show the performance of SepVAE and SepCLR on Digits watermarked on CIFAR-10 and CelebA with

Table 17: Balanced Accuracy results on Digits watermarked on CIFAR-10. Comparison between $\mathcal{L}_{\text{unif}}$ and $\mathcal{L}_{\text{log-sum-exp}}$ to estimate and minimize $H(C)$ and $H(S)$.

|  | DIGITS | | OBJECTS | | $\delta \downarrow$ |
| --- | --- | --- | --- | --- | --- |
|  | S $\uparrow$ | C $\downarrow$ | S $\downarrow$ | C $\uparrow$ | |
| SEPCLR-K-JEM ($\mathcal{L}_{\text{UNIF}}$) | 96.2 | 11.0 | 10.4 | 73.2 | 32.0 |
| SEPCLR-K-JEM (LOG-SUM-EXP) | 96.6 | 11.6 | 11.0 | 71.6 | 34.4 |
| BEST EXPECTED | 100.0 | 10.0 | 10.0 | 100.0 | 0.0 |

Table 18: Balanced Accuracy results on CelebA accessories. Comparison between $\mathcal{L}_{\text{unif}}$ and $\mathcal{L}_{\text{log-sum-exp}}$ to estimate and minimize $H(C)$ and $H(S)$.

|  | DIGITS | | OBJECTS | | $\delta \downarrow$ |
| --- | --- | --- | --- | --- | --- |
|  | S $\uparrow$ | C $\downarrow$ | S $\downarrow$ | C $\uparrow$ | |
| SEPCLR-K-JEM ($\mathcal{L}_{\text{UNIF}}$) | 98.57 | 55.21 | 62.52 | 78.00 | 41.16 |
| SEPCLR-K-JEM (LOG-SUM-EXP) | 98.73 | 55.06 | 61.36 | 76.94 | **40.75** |
| BEST EXPECTED | 100.0 | 10.0 | 10.0 | 100.0 | 0.0 |

accessories with different architectures. SepVAE with ResNet-18 performs less better or similarly than the one described in our original paper. Conversely, SepCLR with ResNet-18 performs better. Overall, SepCLR remains largely better than SepVAE, a consistent method across Contrastive Analysis VAEs.

Table 19: Results of several different encoder architectures on Digits watermarked on CIFAR (B-ACC).

|  | DIGITS | | OBJECTS | | $\delta_{\text{TOT}} \downarrow$ |
| --- | --- | --- | --- | --- | --- |
|  | S $\uparrow$ | C $\downarrow$ | S $\downarrow$ | C $\uparrow$ | |
| SEPVAE | 90.6 | 17.8 | 10.6 | 36.6 | 81.2 |
| SEPVAE - RESNET 18 ENCODER | 90.8 | 23.2 | 10.2 | 34.0 | 88.24 |
| SEPCLR WITH SEPVAE'S ENCODER | 75.6 | 28.8 | 16.2 | 52.6 | 96.8 |
| SEPCLR-K-JEM | 96.2 | 11.0 | 10.4 | 73.2 | **32.0** |
| BEST EXPECTED | 100.0 | 10.0 | 10.0 | 100.0 | 0.0 |

Table 20: Results of several different encoder architectures on CelebA accessories (B-ACC).

|  | HATS/GLSS | | SEX | | $\delta_{\text{TOT}} \downarrow$ |
| --- | --- | --- | --- | --- | --- |
|  | S $\uparrow$ | C $\downarrow$ | S $\downarrow$ | C $\uparrow$ | |
| SEPVAE | 84.46 | 65.19 | 60.12 | 59.20 | 81.65 |
| SEPVAE WITH RESNET-18 ENCODER | 86.13 | 67.47 | 60.04 | 61.93 | 81.45 |
| SEPCLR - K-JEM WITH SEPVAE'S ENCODER | 97.89 | 60.01 | 51.07 | 70.51 | 42.68 |
| SEPCLR - K-JEM | 98.57 | 55.21 | 62.52 | 78.00 | **41.16** |
| BEST EXPECTED | 100.0 | 50.0 | 50.0 | 100.0 | 0.0 |

## G.8 DSPRITES ELEMENT SUPERIMPOSED ON A DIGIT GRID

We show supplementary qualitative results on the salient space disentanglement in Fig. 6. We qualitatively show that the common space captures background variability rather than foreground variability in Fig. 7. We qualitatively show that the salient space captures foreground variability rather than background variability in Fig. 8.

We also show quantitative results in Tab. 21 by computing the Mutual Information Gap (MIG) score to measure the disentanglement in the DSprite-MNIST experiment. Results are reported below, and it can be noticed that the proposed method obtains good results (MIG is bounded by 0 and 1, where 1 indicates a perfect result).

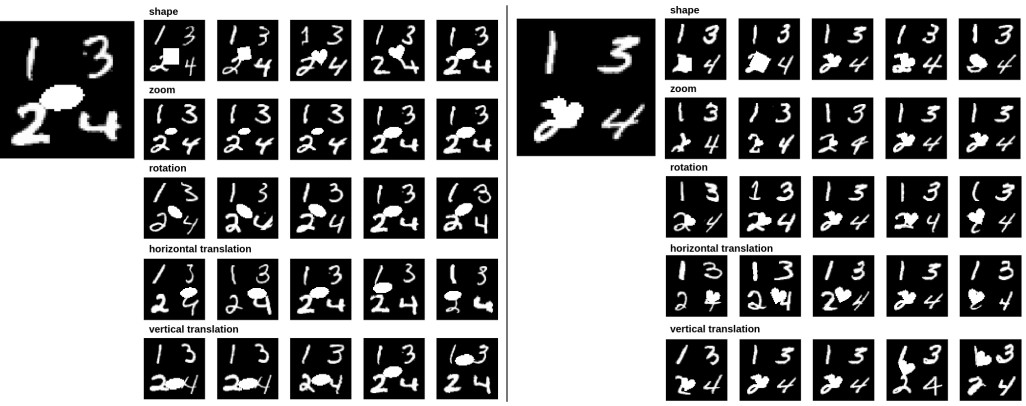

Figure 6: Attribute Supervised SepCLR on dSprites superimposed on a digits grid. Given an image, sampling of the nearest neighbor images in the latent space given small displacement given an axis of the salient space. Each row represents the variation of only one element of the salient factor $s$ while keeping $c$ fixed. We can see a certain disentanglement: shape (line 1), zoom (line 2), orientation (line 3), X and Y position (lines 4 and 5).

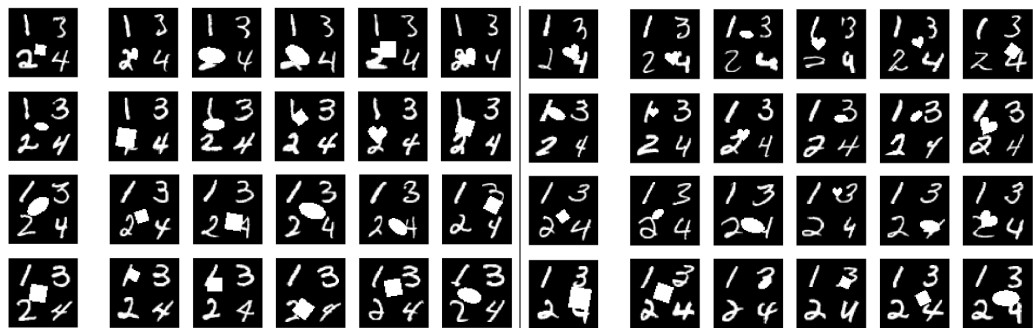

Figure 7: Attribute Supervised SepCLR on dSprites superimposed on a digits grid. Given random target images on the left, we sample the nearest neighbors in the dataset with respect to their L2 distance in the common space only. We can see that the dSprite object remains the same while the MNIST digit grid in the background changes across the neighbors.

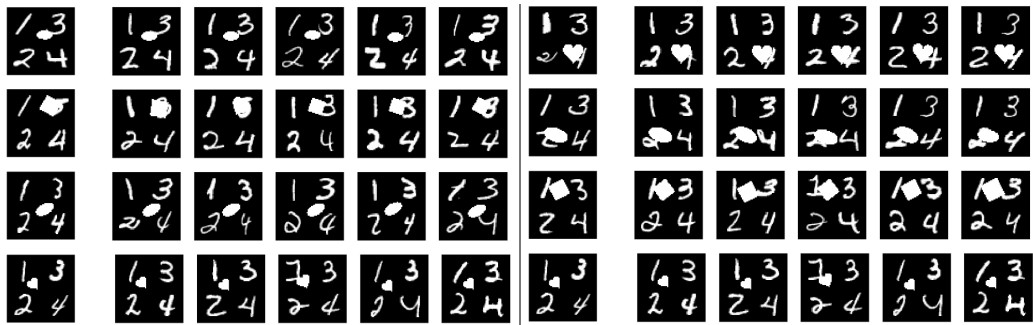

Figure 8: Attribute Supervised SepCLR on dSprites superimposed on a digits grid. Given random target images on the left, we sample the nearest neighbors in the dataset with respect to their L2 distance in the salient space only, we can see that the dSprite object remains the same while the MNIST digits in the background change across the neighbors.

Table 21: Computation of MIG on the salient space in dSprites on MNIST digit grid experiment.

| | Z1 (SHAPE) | Z2 (ZOOM) | Z3 (ROTATION) | Z4 (TRANS X) | Z5 (TRANS Y) | AVG |
|---|---|---|---|---|---|---|
| BEST EXPECTED | 1 | 1 | 1 | 1 | 1 | 1 |
| ATTR SUP SEPCLR - K-JEM | 0.915 | 0.909 | 0.674 | 0.823 | 0.835 | 0.831 |
| RANDOM VECTOR | 0.002 | 0.003 | 0.007 | 0.007 | 0.0008 | 0.003 |

### G.9 QUALITATIVE RESULTS ON CELEBA WITH ACCESSORIES

In this section, we propose to display qualitative results on the CelebA accessories dataset. In Fig. 9 and Fig. 10, we propose to respectively display a 2D t-SNE plot for the salient and common latent space of SepCLR-k-JEM on the target dataset (portraits of celebrities with accessories). Yellow points represent people with hats, Purple points represent people with glasses. We can clearly see that our method has correctly encoded the patterns related to the accessories in the salient space and not in the common space.

Figure 9: 2D t-SNE plot of the salient space of SepCLR-k-JEM on CelebA with accessories. We highlight in yellow and purple the actual labeled subgroups (people with hats or with glasses), respectively. We can see that the two subgroups are clearly separated in the salient space. Furthermore, we train a K-Means (K=2), which successfully identifies the two subgroups, and we propose to display the 6 nearest images from both centroids. Interestingly, we observe various backgrounds, poses, and people of different genders but with the same accessories (hats in cluster 0 and glasses in cluster 1). This clearly shows that our method has correctly encoded the patterns related to the accessories in the salient space and not the general ones (e.g., background, pose, gender, etc.).

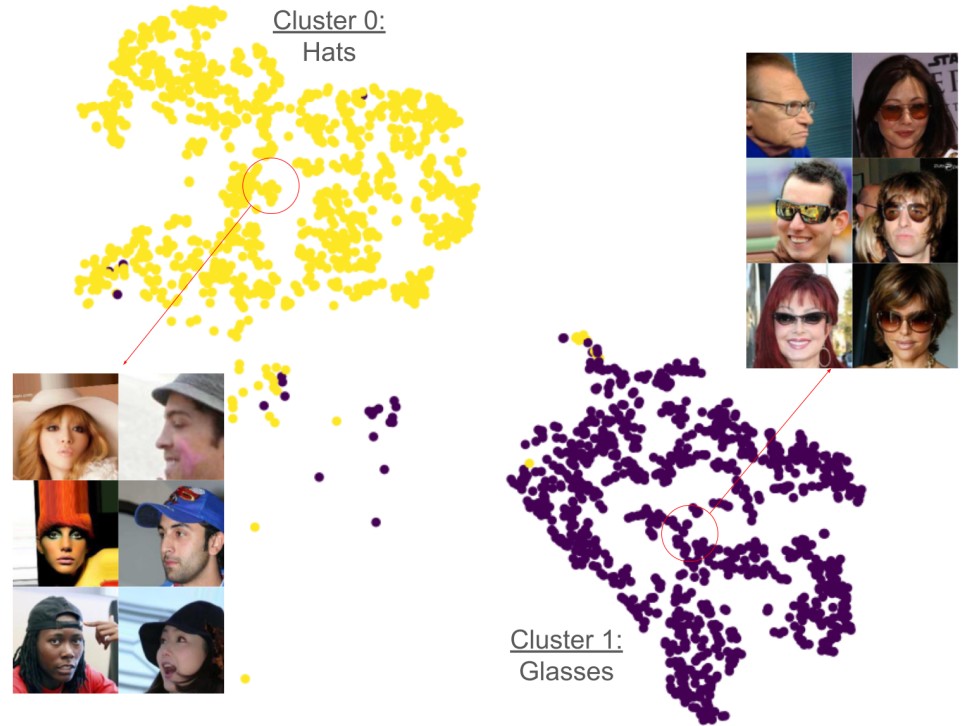

Figure 10: 2D t-SNE plot of the common space of SepCLR-k-JEM on CelebA with accessories (target dataset). We highlight in yellow and purple the actual labeled subgroups (people with hats or with glasses), respectively. We can see that the two subgroups overlap in the common space. This clearly confirms that our method does not encode the patterns related to the accessories in the common space.

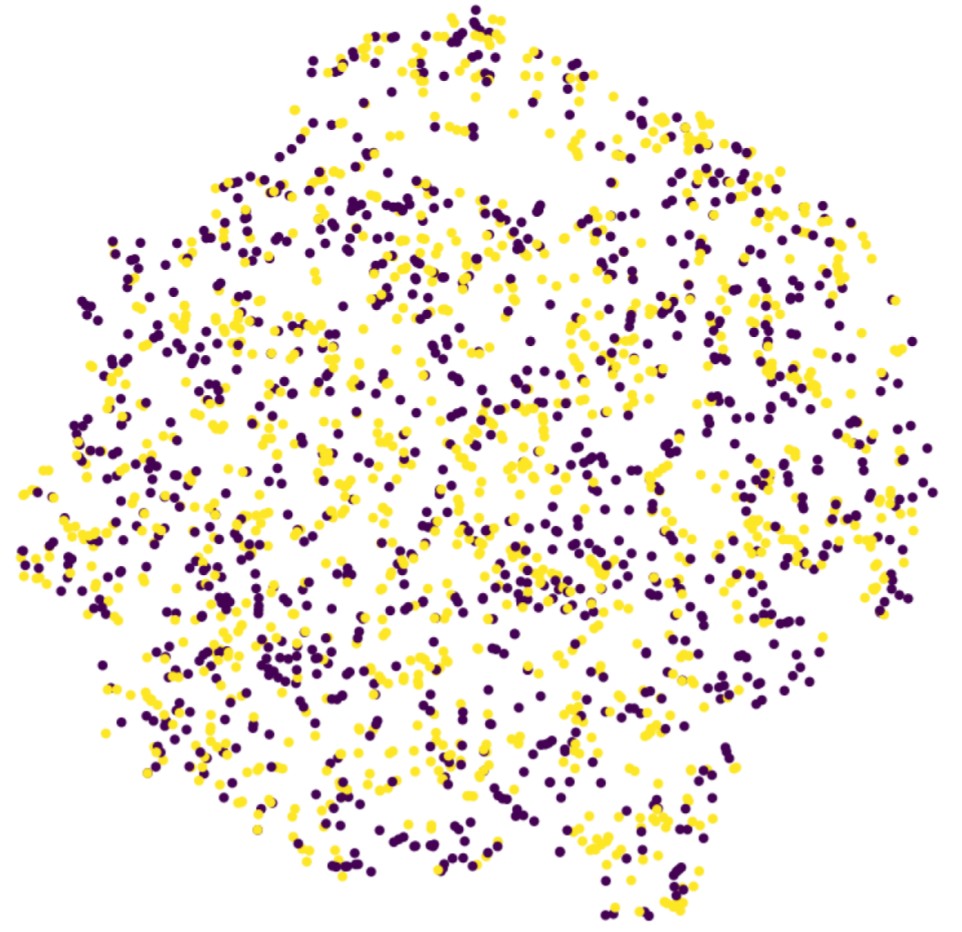

