# OpenReview forum: "Separating common from salient patterns with Contrastive Representation Learning"
_ICLR.cc/2024/Conference — ICLR 2024 poster_

### Official Review · Reviewer_4LjD · 2023-10-29

**Soundness:** 3 good
**Presentation:** 3 good
**Contribution:** 3 good
**Rating:** 8
**Confidence:** 3

**Summary:**

Ths paper discusses Contrastive Analysis, a sub-field of Representation Learning that aims to distinguish common and salient factors of variation between healthy and diseased datasets.
Current models based on Variational Auto-Encoders have shown poor performance in learning semantically expressive representations.
In contrast, Contrastive Representation Learning has shown significant advancements in various applications.
The proposed method, called Sep-CLR, leverages Contrastive Learning to acquire semantically expressive representations suitable for Contrastive Analysis by utilizing the InfoMax Principle and optimizing Mutual Information terms.
The paper provides both theoretical and experimental analysis.

**Strengths:**

1. The theoretical analysis is reasonable and easy to follow.
2. The proposed method outperforms baselines by a significant margin on several datasets.

**Weaknesses:**

1. The submission format of the paper should change to ICLR 2024. It is ICLR 2023 now.
2. It will be better to give some qualitative results on not only the mnist dataset but also X-ray or other real-application data.

**Questions:**

1. It would be better if higher-resolution images could be provided in Figure 1.

---

> ### Author Response · Authors · 2023-11-21
> **Quality of the paper**
>
> Q: It would be better if higher-resolution images could be provided in Figure 1.
> A: We have updated Fig. 1 with a higher-resolution image.
>
> Q:The submission format of the paper should change to ICLR 2024. It is ICLR 2023 now.
> A: We thank the reviewer for pointing this out. We will change the submission format in the revised version of the manuscript.

---

> ### Author Response · Authors · 2023-11-22
> **Qualitative results**
>
> Q: It will be better to give some qualitative results on not only the MNIST dataset but also X-ray or other real-application data.
>
> A: We agree with the Reviewer that it would be interesting for the reader to have some qualitative results. However, we could add an easy-to-interpret qualitative result only for CelebA (see Sec. G.9 of Supplementary). Indeed, the three clinical applications are very challenging, and the visual differences between clinical subtypes are very subtle and difficult to see with the naked eye, even for expert clinicians. We could not find so far satisfying qualitative results that were interpretable and easy to understand (for non-experts). We have asked our medical colleagues to help us understand the qualitative results and find suitable visual examples, but they did not have time in the past week. They should be available by the end of this week, and we hope we will be able to add interesting and legible qualitative results, for at least one clinical application, to the final version of the manuscript.

---

### Official Review · Reviewer_F95G · 2023-10-29

**Soundness:** 3 good
**Presentation:** 2 fair
**Contribution:** 2 fair
**Rating:** 5
**Confidence:** 2

**Summary:**

The paper proposes a novel theoretical framework for Contrastive Analysis based on the InfoMax principle, leveraging Contrastive Learning to estimate the common and salient terms, and suggests a strategy to reduce the information leakage between the common and salient spaces. Specifically, the framework consists of two InfoMax terms for the common space and the salient space, and k-JEM for preventing information leakage. In addition, the authors propose a Supervised InfoMax term to disentangle the salient factors.

The key contributions are:

1) Reformulating Contrastive Analysis under the InfoMax principle with two Mutual Information terms to maximize - one for common factors and one for salient factors unique to the target dataset.

2) Leveraging Contrastive Learning losses to estimate these Mutual Information terms - retrieving InfoNCE for the common factors and proposing a new background-contrasting loss for the salient factors.

3) Introducing a new strategy called k-JEM to maximize joint entropy for reducing information leakage between common and salient spaces.

4) Extending the framework with a Supervised InfoMax term to disentangle salient factors when attributes are available.

The experimental results on 5 datasets show k-JEM outperforms other mutual information minimization techniques and significantly promotes separating salient factors and common factors.

Overall, the proposed SepCLR framework and k-JEM regularization demonstrate strong empirical results for contrastive analysis on both visual and medical datasets.

**Strengths:**

Here are some strengths of the paper:

1. The proposed theoretical framework provides new insights into Contrastive Analysis by formulating it under the InfoMax principle and identifying key mutual information terms to estimate. This enlightens future work on estimating these terms for contrastive analysis.

2. The paper proposes a strategy to disentangle target-specific attributes within the salient space in a supervised manner when attributes are available. This extends the framework's capabilities.

3. The paper provides an extensive discussion and comparison of several mutual information variational upper bound methods (vCLUB, vUB, vL1out, TC) as well as the strategies of mutual information minimization and distribution matching for reducing information leakage.

4. The derivation of the InfoNCE loss and its alignment and uniformity terms from the InfoMax principle is clearly explained, connecting contrastive learning and information theory foundations.

**Weaknesses:**

1.  The choice of metrics could also be expanded and analyzed in more detail. For example, the reasoning behind expecting certain accuracy scores is not fully clear. Furthermore, it is not comparable between accuracy of 0% and 20% for (digits,C) on CIFAR-10.

2. The evaluation is limited to a small set of datasets and tasks. A more comprehensive evaluation on a wider variety of datasets and downstream tasks could strengthen the results. There is limited discussion of hyperparameter sensitivity and scalability to larger datasets. Analyzing the impact of key hyperparameters and demonstrating scalability would be useful.

3. In the disentanglement experiment (Figure 2), some entanglement seems to remain between factors. Using quantitative metrics like MIG and DCI could help analyze this. The sprite changes in Figure 2(b) could also be explained.

4. Some architectural and mathematical details are unclear:
- The encoder architectures and whether they are independent could be specified.
- The notation for views v and number of samples Nx, Ny could be clarified.
- Formulas and descriptions could be expanded for readability.

**Questions:**

Here are some potential questions about the paper:

1. How does Equation 7 constrain `s'` to be information-less? This equation seems to promote the embeddings to be uniformly distributed rather than constraining s' specifically. Some clarification on how the information-less hypothesis is enforced would be helpful.

2. The alignment terms in Equations 4 and 6 look different - one uses a log summation inside the log, while the other does not. What is the reason for this difference in formulations between the common space alignment (Eq 4) and salient space alignment (Eq 6)? Some explanation or intuition here could help the reader understand.

---

> ### Author Response · Authors · 2023-11-21
> **Authors response to Weakness 1**
>
> Q: The choice of metrics could also be expanded and analyzed in more detail. For example, the reasoning behind expecting certain accuracy scores is not fully clear. Furthermore, it is not comparable between the accuracy of 0\% and 20\% for (digits, C) on CIFAR-10.
>
> A: To quantify the ability of a method to separate common from target-specific variability factors, we train a linear classifier (or regressor) on the inferred salient and common representations of the test samples. In this way, we can assess whether the information about an attribute, which is present either in both datasets or only in the target dataset, is indeed captured in the common or  salient latent space. For instance, taking as example the CIFAR10-MNIST dataset, a model should successfully capture the background variability (i.e., CIFAR objects) only in the common latent space, and the MNIST variability (i.e., digits) only in the salient space. Thus, a perfect classifier would have 100 \% of balanced accuracy on MNIST when using "S" and 10\% (which corresponds to randomness) when using "C" . Conversely, it should have 100\% of balanced accuracy on Cifar-10 when trained on "C" and 10\% when trained on "S". We add the following sentence for clarity in the revised manuscript: ”"Best Expected" reports a perfect results (100) when the attribute should be present in that latent space and a random result (50) when it should not."

---

> ### Author Response · Authors · 2023-11-21
> **Authors response to Weakness 2**
>
> Q: The evaluation is limited to a small set of datasets and tasks. A more comprehensive evaluation on a wider variety of datasets and downstream tasks could strengthen the results. There is limited discussion of hyperparameter sensitivity and scalability to larger datasets. Analyzing the impact of key hyperparameters and demonstrating scalability would be useful.
>
> A: In our paper, we used $5$ different datasets for traditional Contrastive Analysis and $1$ dataset for Contrastive Analysis with independent fine-grained attributes on the target dataset. It is, to the best of our knowledge, the Contrastive Analysis paper with the largest experimental setup in the literature. Also, it is important to note that there is no consensual benchmark to evaluate this kind of methods, which makes difficult the evaluation on a wider variety of datasets. Concerning the hyperparameters, we perform a sensitivity analysis on the k-CEM strength in Tab. 6, Sec G. 1, and provide an Ablation Study as asked by Reviewer fALd, see  Tables 13 and 14 of the revised manuscript, Sec.G4.
>
> Table 13: Ablation Study on Digits watermarked on CIFAR-10 (B-ACC).
>
> |                                              | Digits       | Digits         | Objects        | Objects      | $\delta_\text{tot} \downarrow$ |
> |----------------------------------------------|--------------|----------------|----------------|--------------|--------------------------------|
> |                                              | S $\uparrow$ | C $\downarrow$ | S $\downarrow$ | C $\uparrow$ |                                |
> | SepCLR-$\lambda=0$ (no k-JEM)                | 95.6         | 94.4           | 9.0            | 42.0         | 145.8                          |
> | SepCLR-$\beta=0$ (no Infoless reg)           | 96.2         | 11.6           | 10.4           | 71.8         | 34.0                           |
> | SepCLR-$\lambda_S=0$ (no Salient term)       | 93.4         | 42.0           | 18.6           | 40.0         | 90.25                          |
> | SepCLR-$\lambda_C=0$ (no Common term)        | 94.4         | 10.4           | 18.8           | 20.4         | 94.4                           |
> | SepCLR-k-JEM                                 | 96.2         | 11.0           | 10.4           | 73.2         | 32.0                           |
> | Best expected                                | 100.0        | 10.0           | 10.0           | 100.0        | 0.0                            |
>
>
> Table 14: Ablation Study on Digits watermarked on CelebA with accessories (B-ACC).
> |                                              | Hats/Glss    | Hats/Glss      | Sex            | Sex          | $\delta_\text{tot} \downarrow$   |
> |----------------------------------------------|--------------|----------------|----------------|--------------|--------------------------------|
> |                                              | S $\uparrow$ | C $\downarrow$ | S $\downarrow$ | C $\uparrow$ |                                |
> | SepCLR - $\lambda=0$ (no k-JEM)              | 99.03        | 66.68          | 98.48          | 79.48        | 86.65                          |
> | SepCLR - $\beta=0$ (no Infoless reg)         | 99.12        | 53.88          | 68.82          | 77.29        | 46.29                          |
> | SepCLR - $\lambda_S=0$ (no Salient term)     | 77.50        | 87.73          | 53.30          | 77.55        | 85.98                          |
> | SepCLR - $\lambda_C=0$ (no Common term)      | 98.38        | 56.32          | 66.44          | 53.09        | 71.29                          |
> | SepCLR - k-JEM                               | 98.57        | 55.21          | 62.52          | 78.00        | 41.16                          |
> | Best Expected                                | 100.0        | 50.0           | 50.0           | 100.0        | 0.0                            |

---

> ### Author Response · Authors · 2023-11-21
> **Authors response to Weakness 3**
>
> Q: In the disentanglement experiment (Figure 2), some entanglement seems to remain between factors. Using quantitative metrics like MIG and DCI could help analyze this. The sprite changes in Figure 2(b) could also be explained'
>
> A:  As suggested by the Reviewer, we used the Mutual Information Gap (MIG) score [1] to measure the disentanglement in the DSprite-MNIST experiment. Results are reported in Sec G.8 of the revised manuscript., and it can be noticed that the proposed method obtains good results (MIG is bounded by 0 and 1, where 1 indicates a perfect result).
>
> Table 21: Computation of MIG on the salient space in dSprites on MNIST digit grid experiment.
> |                         | Z1 (Shape) | Z2 (Zoom) | Z3 (Rotation) | Z4 (Trans X) | Z5 (Trans Y) | Avg   |
> |-------------------------|------------|-----------|---------------|--------------|--------------|-------|
> | Best Expected           | 1          | 1         | 1             | 1            | 1            | 1     |
> | Attr Sup SepCLR - k-JEM | 0.915      | 0.909     | 0.674         | 0.823        | 0.835        | 0.831 |
> | Random Vector           | 0.002      | 0.003     | 0.007         | 0.007        | 0.0008       | 0.003 |
>
>
> [1] Ricky T. Q. Chen et al., Isolating Sources of Disentanglement in VAEs, NeurIPS, 2018.
>
> Concerning Fig. 2, we showed the image in the dataset whose inferred common latent vector is the closest in terms of L2 distance with no constraint on its salient space. Therefore, we expect to retrieve images with a similar background (MNIST digits with a similar style, rotation, boldness, etc...) but with no constraint on the dSprite element (because the common space is expected to capture background factors of variability, not dSprite related factors of variability).

---

> ### Author Response · Authors · 2023-11-21
> **Authors response to Weakness 4**
>
> Q: The encoder architectures and whether they are independent could be specified.
>
> A: We apologize if we have not been clear enough. The common and salient encoders do not share weights but have the same architecture.

---

> ### Author Response · Authors · 2023-11-21
> **Authors response to Weakness 5**
>
> Q: How does Equation 7 constrain $s'$ to be information-less? This equation seems to promote the embeddings to be uniformly distributed rather than constraining s' specifically. Some clarification on how the information-less hypothesis is enforced would be helpful.
>
> A:  There is a misunderstanding here, and we apologize if our original formulation was not clear. In our method, $s'$ is constant, and set to the null vector in practice: $s'=0$. Eq. 7 does not constrain $s'$ to be information-less. We actually enforce the salient factors $s$ of the background dataset to be always equal to the constant value $s'$, and thus information-less, by minimizing the third term of Eq. 8, i.e: the Lagrangian regularization term :
>
> $- \beta \frac{1}{N_X} \sum_{i=1}^{N_X} \frac{||f_{\theta_s}(t(x_i)) - s'||_2^2}{2 \tau}$
>
> which basically minimizes the distance between salient representations of background samples $f_{\theta_s}(t(x_i))$ and $s'$. Differently, Eq. 7's first term encourages the salient representations of the target samples $f_{\theta_s}(t(y_i))$ to be repelled from the information-less vector $s'$.

---

> ### Author Response · Authors · 2023-11-21
> **Authors response to Weakness 6**
>
> A: The alignment terms in Equations 4 and 6 looks different - one uses a log summation inside the log, while the other does not. What is the reason for this difference in formulations between the common space alignment (Eq. 4) and salient space alignment (Eq. 6)? Some explanation or intuition here could help the reader understand.
>
> Q: We report here the same answer given to Reviewer etAa's Weakness 2: https://openreview.net/forum?id=30N3bNAiw3&noteId=2vmzO0P4un

---

> ### Author Response · Authors · 2023-11-21
> **Readability**
>
> Q: Formulas and descriptions could be expanded for readability. The notation for views v and number of samples Nx, Ny could be clarified.
>
> A: We apologize if these points were not clear. We have expanded the formulas and clarified the notations in the revised version of the manuscript.

---

### Official Review · Reviewer_etAa · 2023-10-30

**Soundness:** 3 good
**Presentation:** 3 good
**Contribution:** 2 fair
**Rating:** 8
**Confidence:** 4

**Summary:**

This paper proposes a technique called SepCLR that uses contrastive learning in order to learn representations that separate common elements from salient ones for the downstream task at hand. The authors examine the performance of their method in creating separate representations for these two elements, and demonstrate improvements over previous work on the same subject.

**Strengths:**

- The method proposed by the authors is novel, as far as I am aware. Limiting the shared information between the common and the salient space explicitly is an interesting way to ensure that the two encoders model different aspects of the data, leading to less overlap in the information between the two encoders.

- The authors have performed extensive experiments on a variety of datasets, and have also examined several different variations of their proposed method, as can be seen in Table 1. I also appreciate the fact that the datasets used are not only standard ones like MNIST or CIFAR-10, but also come from the medical domain (although I would also appreciate results involving more complicated datasets like CIFAR-100 or ImageNet, which have more object classes available).

- I also appreciate the detailed analysis that the authors provide for the datasets used in the appendix.

**Weaknesses:**

- As far as I understand, there is an inherent limitation for the method in that knowing the labels for the target dataset is required during training. This limits the applicability of SepCLR in the unsupervised setting, which is also the one most commonly examined by contrastive learning works.

- I believe that there are some issues with the proposed method, that I would be grateful if the authors could elaborate on:

  - The authors make some decisions when designing the loss that go against what is commonly done in related contrastive learning papers. In particular, the loss they propose has the formulation of $L_{unif}$ as found in Wang & Isola [A], but the most commonly used formulation is that of InfoNCE, which differs in that the resulting loss is a sum of Log-Sum-Exp functions, instead of a single Log-Sum-Exp. Similarly, in the alignment term they use a formulation closer to $L_{out}$ from Supervised Contrastive Learning [B], but the same paper notes that another formulation that simply sums the inner products, named $L_{in}$, is better experimentally (the authors examine this in the appendix, but do not explain why they chose $L_{out}$). I would be grateful if the authors could elaborate on these design decisions.

  - Related to the above, it seems that the alignment terms in the common space and in the salient space are different (and similar to $L_{out}$ and $L_{in}$ respectively). I would be glad if the authors could explain why this is the case.

  - In Equation (7), the first term in the sums essentially forces the representations of the salient encoder to be far from the constant vector $s’$. It’s not immediately clear to me why this term is there - it doesn’t seem to arise from optimizing $\hat{H}(S)$, and the informationless hypothesis only comes into play in Equation (8). I think the authors need to explain this part a bit more.

  - Finally, the zero mutual information constraint is somewhat misleading - I understand the point the authors make that minimizing $I(c;s)$ is not the best thing to do, but at the same time, the proposed method does not directly force $I(c; s) = 0$. There is no guarantee that maximizing $H(c,s)$ does not affect the maximization of $H(c) + H(s)$, nor that the final solution will have $H(c,s) = H(c) + H(s)$. I believe that the authors should be clearer about this point.

- I also believe that some points regarding the presentation of the paper can be improved:

  - Tables 1 and 2 contain several variants of SepCLR, but it is not clear what each of them signify. The authors should better explain the variants of SepCLR in this table.

  - Section 4 seems out of place, as it does not come up later in the main paper, and is in fact extremely similar to Section E in the Appendix. I believe that this part should be moved away from the main paper, as currently it throws the reader off (despite the paper having good structure overall).

- Finally, I believe that it would be good to include the baseline of simply training the model using the entirety of the dataset via e.g. SimCLR. While I’m fairly sure that this will not perform as well, it’s still something good to include to get a sense of why the two different encoders are necessary.

**Questions:**

I would be grateful if the authors could clarify the points I made above regarding the design decisions made for the method and the details of its formulation.

---

> ### Author Response · Authors · 2023-11-21
> **Authors response to Weakness 1**
>
> Q: As far as I understand, there is an inherent limitation for the method in that knowing the labels for the target dataset is required during training. This limits the applicability of SepCLR in the unsupervised setting, which is also the one most commonly examined by contrastive learning works.
>
> A: We apologize if we have not been clear on this specific point. SepCLR's objective relates to Contrastive Analysis (not to be mistaken with Contrastive Learning) and it has not been designed for the unsupervised setting, as it's the case for several contrastive learning works, but for the binary or weakly-supervised setting. Indeed,  we require the knowledge of the binary labels (target/background) for each sample, but not of the patterns of variation specific to the target or background dataset (hence the weakly connotation). Our goal is to discover the underlying generative factors that 1) distinguish a target dataset from a background dataset (i.e., salient factors) and that 2) are shared between them (i.e., common factors).
>
> Please note that, as written in our paper, the ability to distinguish and separate common from salient generative factors given two datasets (a background and a target dataset) is crucial in various domains. For instance, in medical imaging, researchers seek to identify pathological patterns in a population of patients (target) compared to healthy controls (background) Antelmi et al. (2019); Aglinskas et al.(2022). Contrastive Analysis also concerns other domains like drug research (medicated vs. placebo populations), surgery (pre-intervention vs. post-intervention groups), time series (signal vs. signal-free samples), biology and genetics (control vs. characteristic-trait population, Jones et al. (2021) ).
> Finally, please note that at inference, one could also associate the proposed method with a binary classification algorithm to infer the binary class of the test samples. In this way, one could also use our method on an unlabeled test set. However, this is out of the scope of this paper.

---

> > ### Author Response · Authors · 2023-11-21
> > **Authors response to Weakness 3**
> >
> > Q: In Equation (7), the first term in the sums essentially forces the representations of the salient encoder to be far from the constant vector $s'$. It’s not immediately clear to me why this term is there - it doesn’t seem to arise from optimizing $H(s)$, and the information-less hypothesis only comes into play in Equation (8). I think the authors need to explain this part a bit more.
> >
> > A: We apologize if this part was not clear. As shown in Sec. B.2 of the Supplementary, the information-less hypothesis is actually taken into account also in Eq. 7, since we assume that the salient encoder produces background embeddings equal to the information-less vector $s'$. This hypothesis is then explicitly optimised in Eq. 8 with a regularisation term. One term thus enforces the salient encoder to produce background embeddings aligned on the information-less vector $s'$, while the other one forces the representations of the target samples produced by the salient encoder to be far away from $s'$ (and thus background embeddings), as correctly noticed by the Reviewer. We have clarified this point in the revised manuscript at page. 6 by adding:
> >
> > "Using this hypothesis in the computations (see Sec. B.2 of the Supplementary) and ignoring constant terms, we obtain:"

---

> ### Author Response · Authors · 2023-11-21
> **Authors response to Weakness 2**
>
> Q: The authors make some decisions when designing the loss that go against what is commonly done in related contrastive learning papers. In particular, the loss they propose has the formulation of $L_{unif}$ as found in Wang et Isola [A], but the most commonly used formulation is that of InfoNCE, which differs in that the resulting loss is a sum of Log-Sum-Exp functions, instead of a single Log-Sum-Exp. Similarly, in the alignment term they use a formulation closer to $L_\text{out}$ from Supervised Contrastive Learning [B], but the same paper notes that another formulation that simply sums the inner products, named $L_\text{in}$, is better experimentally (the authors examine this in the appendix, but do not explain why they chose $L_\text{out}$). I would be grateful if the authors could elaborate on these design decisions.
> Related to the above, it seems that the alignment terms in the common space and in the salient space are different (and similar to and respectively). I would be glad if the authors could explain why this is the case.
>
> A: As shown in Sec. A.1 of the Supplementary, we estimate the entropy using a resubstitution entropy estimator. This results in one of the terms of the standard Contrastive loss (i.e., InfoNCE) that accounts for the negative samples. As shown in Wang et Isola, this term has the same minimizer as the $L_\text{unif}$ loss when the number of negatives tends to be infinite. We decided to use the $L_\text{unif}$ loss instead of the Contrastive loss because it is computationally less expensive, and it has been shown by Wang et Isola to lead to good representations and good downstream task performance. Furthermore, we have also compared the two losses in the CIFAR10-MNIST dataset the CelebA accessories (see Tables 17 and 18 of the revised manuscript) and found that the results are slightly better or similar using $L_\text{unif}$. These results will be added in the Supplementary Sec G.6 of the revised manuscript.
>
>
> Now, concerning the comparison with $\mathcal{L}^{\text{in}}$ and $\mathcal{L}^{\text{out}}$ introduced in SupCon (Khosla et al, 2021), we apologize if we have not been clear on this specific point. The alignment terms in both spaces have been derived in the same way, using a re-substitution (conditional) entropy estimator (see Sec. A.2 and B.1 in Supplementary). The resulting loss is indeed similar to the $L_\text{sup}^\text{in}$ loss of SupCon (except that, in our formulation, positive samples are views from the same image and not from the same class). However, as specified in the original article, to reduce the computational burden and have a fair comparison with concurrent works, we use in practice only a single view (K=1). This simplifies the loss into the alignment term described in Wang et Isola, 2020 (which is also the numerator of InfoNCE). The two alignment terms of our article (Eq. 4 and Eq. 6 of the original manuscript) can thus be written:
>
> $$  L_{\text{align}} = - \frac{1}{N_X + N_Y} \sum_{i=1}^{N_X + N_Y} \log \frac{1}{K} \sum_{k=1}^K \exp \frac{- || f_{\theta_C}(v_i) - f_{\theta_C}(v_i^k) ||_2^2}{2 \tau} + \log(\sqrt{2 \pi \tau}) $$
>
> $$   L_{\text{align}} \approx  \frac{1}{N_X + N_Y} \sum_{i=1}^{N_X + N_Y} \frac{|| f_{\theta_C}(v_i) - f_{\theta_C}(v_i^+) ||_2^2}{2 \tau}  $$
>
> $$ L_{\text{y-align}} = - \frac{1}{N_Y} \sum_{i=1}^{N_Y} \log \frac{1}{K} \sum_{k=1}^K \exp \frac{- || f_{\theta_C}(t(y_i)) - f_{\theta_C}(t(y_i)^k) ||_2^2}{2 \tau} + \log(\sqrt{2 \pi \tau}) $$
>
> $$ L_{\text{y-align}} \approx \frac{1}{N_Y} \sum_{i=1}^{N_Y} \frac{|| f_\theta(t(y_i)) -  f_\theta(t(y_i)^+||_2^2}{2 \tau} $$
>
> In the original article, we reported in Eq. 6 only the simplified formula (with $K=1$), which might be misleading. We apologize for the inconvenience and propose to keep the full computations in the revised version of the manuscript.
>
> N.B.: If we considered multiple views (i.e.: keeping $K\geq2$), they would be averaged within the $\log$. However, we notice that current multi-view unsupervised contrastive learning methods, like Khosla et al, 2021 or Caron et al, 2021, consistently reported better results when averaging the views outside of the log (e.g., the multi-crop augmentation introduced in Caron et al, 2021). This could be obtained also in our formulation by using Jensen inequality and pulling the sum over $K$ outside of the $\log$ (exactly as for $\mathcal{L}^{\text{in}}$ and $\mathcal{L}^{\text{out}}$ in SupCon (Khosla et al, 2021)). Finding the best way to add multiple views in our framework is an interesting research direction and is left as future work.

---

> > ### Author Response · Authors · 2023-11-22
> > **Tables 19 et 20.**
> >
> > Table 19: Balanced Accuracy results on Digits watermarked on CIFAR-10.
> > Comparison between $L_\text{unif}$ and $L_\text{log-sum-exp}$ to estimate and minimize $H(C)$ and $H(S)$
> >
> > |                            | Digits       | Digits         | Objects        | Objects      | $\delta \downarrow$ |
> > |----------------------------|--------------|----------------|----------------|--------------|---------------------|
> > |                            | S $\uparrow$ | C $\downarrow$ | S $\downarrow$ | C $\uparrow$ |                     |
> > | SepCLR-k-JEM (L unif)      | 96.2         | 11.0           | 10.4           | 73.2         | 32.0                |
> > | SepCLR-k-JEM (log-sum-exp) | 96.6         | 11.6           | 11.0           | 71.6         | 34.4                |
> > | Best expected              | 100.0        | 10.0           | 10.0           | 100.0        | 0.0                 |
> >
> > Table 20: Balanced Accuracy results on CelebA accessories.
> > Comparison between $L_\text{unif}$ and $L_\text{log-sum-exp}$ to estimate and minimize $H(C)$ and $H(S)$.
> > |                            | Hats/Glsses  | Hats/Glsses    | Sex            | Sex          | $\delta \downarrow$ |
> > |----------------------------|--------------|----------------|----------------|--------------|---------------------|
> > |                            | S $\uparrow$ | C $\downarrow$ | S $\downarrow$ | C $\uparrow$ |                     |
> > | SepCLR-k-JEM (L unif)      | 98.57        | 55.21          | 62.52          | 78.00        | 41.16               |
> > | SepCLR-k-JEM (log-sum-exp) | 98.73        | 55.06          | 61.36          | 76.94        | 40.75      |
> > | Best expected              | 100.0        | 10.0           | 10.0           | 100.0        | 0.0                 |

---

> ### Author Response · Authors · 2023-11-21
> **Authors response to Weakness 4**
>
> Q: Finally, the zero mutual information constraint is somewhat misleading - I understand the point the authors make that minimizing $I(c, s)$ is not the best thing to do, but at the same time, the proposed method does not directly force $I(c, s) = 0$. There is no guarantee that maximizing does not affect the maximization of $H(c) + H(s)$, nor that the final solution will have $H(c, s) = H(c) + H(s)$.
>
> A: In practice, the maximization of $H(c, s)$ is led simultaneously with the maximization of $H(c)$ and $H(s)$. Indeed, the Common and Salient InfoMax losses seek to maximize the two individual entropy terms.
> Furthermore, it's interesting to notice that since the entropy of a discrete random variable is always upper bounded by the log of the range of the variable and $H(c, s) \leq H(c) + H(s)$ is always verified, maximizing $H(c, s)$, $H(c)$ and $H(s)$ simultaneously should bring to a maximum at which the condition $H(c, s) = H(c) + H(s)$ is fulfilled and thus also $I(c, s) = 0$.
>
> To clarify this point, we modified the article writing in Sec. 3.3 "Furthermore, the Common and Salient InfoMax losses of our framework seek to maximize $H(c)$ and $H(s)$ rather than..." .. " we propose to simultaneously maximize H(c,s), H(c) and H(s), until..."

---

> ### Author Response · Authors · 2023-11-21
> **Authors response to Weakness 5**
>
> Q: Tables 1 and 2 contain several variants of SepCLR, but it is not clear what each of them signify. The authors should better explain the variants of SepCLR in this table.''
>
> A: In Tables 1 and 2 of the original manuscript, we investigated different Mutual Information minimization strategies in order to demonstrate that ours (k-JEM) is the best. They all described in Supplementary Materials Sec.F7. First, we reported the results when minimizing the entropies $H(C)$ and $H(S)$ (even though we argued that it is not optimal in the original paper). We entitled this experiment "k-MI", because it is like minimizing the full Mutual Information (kernel-MI), estimated with a kernel density estimator, rather than only the Joint Entropy (kernel-JE Min), also estimated with a kernel density estimator. Second, we reported the results when minimizing the Total Correlation as described in SepVAE (Louiset et al, 2022) (TC). Then, we reported the results when minimizing the Maximum Mean Discrepancy (MMD) in the common space between background and target samples, as in Weinberger et al, 2022, and we also performed a sensitivity analysis in Sec.G.1. Ultimately, we compared with three MI minimization strategies: CLUB, VUB, and L1out estimated in a variational manner, as described in the paper CLUB (Cheng et al, 2020). Importantly, these methods require training an independent estimator through an asymmetric setting (either by estimating $C$ from $S$ or vice-versa). In order to be thorough, we investigated the use of these methods when estimating the Mutual Information by inferring $C$ from $S$ ($S \rightarrow C$), or $S$ from $C$ ($C \rightarrow S$), or with the average of both cases (sym: symmetric setting). A detailed description of these variants is presented in Supplementary Materials Sec.F7. We added the sentence "See Sec.F7 for details." to the caption of the Table to clarify this point.

---

> ### Author Response · Authors · 2023-11-21
> **Authors response to Weakness 6**
>
> Q: Section 4 seems out of place, as it does not come up later in the main paper, and is in fact extremely similar to Section E in the Appendix. I believe that this part should be moved away from the main paper, as currently, it throws the reader off (despite the paper having good structure overall).
>
> A: Section E in the Supplementary is indeed very similar to Sec. 4 of the original paper and it will be removed in the revised manuscript. This was a mistake and we apologize for that.
>
> However, this part does come up later in the paper and, more exactly, in Fig. 2 where we disentangle the dSprite dataset while contrasting it with a novel background dataset. We believe that this is an interesting research avenue and we would like to keep it in the main paper where we would also add additional quantitative results, as suggested by reviewer F95G.

---

> ### Author Response · Authors · 2023-11-21
> **Authors response to Weakness 7**
>
> Q: Finally, I believe that it would be good to include the baseline of simply training the model using the entirety of the dataset via e.g. SimCLR. While I’m fairly sure that this will not perform as well, it’s still something good to include to get a sense of why the two different encoders are necessary.
>
> A: We agree with the Reviewer that this is an interesting baseline. Additionally , we have also tested a combination of SimCLR and SupCon to have a more fair comparison. We trained SimCLR using only the background dataset and SupCon was trained to discriminate between the background and target samples. In this way, the features learnt by SimClr should only describe the common part while the SupCon's features should only capture the target-specific patterns. As it can be seen from the Tables 10, 11, 12 of the revised manuscript  (added in the Supplementary Sec. G.3), our method clearly outperforms the other two baselines.
>
>
> Table 10: Comparison of SepCLR-k-JEM with Contrastive methods on Digits watermarked on CIFAR-1 (B-ACC)
>
> |                     | Digits       | Digits         | Objects        | Objects      | $\delta_\text{tot} \downarrow$ |
> |---------------------|--------------|----------------|----------------|--------------|--------------------------------|
> |                     | S $\uparrow$ | C $\downarrow$ | S $\downarrow$ | C $\uparrow$ |                                |
> | SimCLR on TG and BG | 44.0         | 44.0           | 94.6           | 94.6         | 180.0                          |
> | SimCLR + SupCon     | 41.4         | 51.4           | 19.0           | 50.0         | 159.0                          |
> | SepCLR-k-JEM        | 96.2         | 11.0           | 10.4           | 73.2         | 32.0                           |
> | Best expected       | 100.0        | 10.0           | 10.0           | 100.0        | 0.0                            |
>
>
>
> Table 11: Comparison of SepCLR-k-JEM with Contrastive methods on CelebA accessories (B-ACC).
>
> |                     | Hats/Glss    | Hats/Glss      | Sex            | Sex          | $\delta_\text{tot} \downarrow$ |
> |---------------------|--------------|----------------|----------------|--------------|--------------------------------|
> |                     | S $\uparrow$ | C $\downarrow$ | S $\downarrow$ | C $\uparrow$ |                                |
> | SimCLR on BG and TG | 98.92        | 98.92          | 84.16          | 84.16        | 100.0                          |
> | SimCLR + SupCon     | 97.93        | 82.15          | 59.98          | 80.76        | 63.44                          |
> | SepCLR-k-JEM        | 98.57        | 55.21          | 62.52          | 78.00        | 41.16                |
> | Best Expected       | 100.0        | 50.0           | 50.0           | 100.0        | 0.0                            |
>
>
>
> Table 12: Comparison of SepCLR-k-JEM with Contrastive methods on ODIR dataset (B-ACC).
>
> |                     | Subtype      | Subtype        | Sex            | Sex          | $\delta_\text{tot} \downarrow$ |
> |---------------------|--------------|----------------|----------------|--------------|--------------------------------|
> |                     | S $\uparrow$ | C $\downarrow$ | S $\downarrow$ | C $\uparrow$ |                                |
> | SimCLR on BG and TG | 66.10        | 66.10          | 57.20          | 57.20        | 125.0                          |
> | SimCLR + SupCon     | 68.70        | 57.17          | 51.94          | 58.41        | 107.0                          |
> | SepCLR-k-JEM        | 68.54        | 47.71          | 52.48          | 59.62        | 97.03                 |
> | Best Expected       | 100.0        | 25.0           | 50.0           | 100.0        | 0.0                            |

---

> ### Comment · Reviewer_etAa · 2023-11-22
> **Response to author rebuttal.**
>
> I'd like to thank the authors for the detailed response to my comments. While most of my concerns have been addressed, I'd like to point out one final thing regarding Weakness 4. While I understand that maximizing all three of $H(c,s)$, $H(c)$ and $H(s)$ should give $I(c, s) = 0$, unless I am mistaken this contains the implicit assumption that the encoders for $c$ and $s$ can model any distribution. I think having the assumption is fine in the theory of the paper, but it should be stated explicitly.
>
> Nevertheless, I still lean towards accepting the paper (since the above point can be clarified in the text).

---

> > ### Author Response · Authors · 2023-11-22
> > **Adding implicit assumption**
> >
> > We kindly thank the Reviewer for his/her positive comment.
> > We agree with her/him about the implicit assumption.
> >
> > We have thus modified the final manuscript adding as footnote at page. 6 : "In this work, we implicitly assume that the encoders $f_{\theta_c}$ and $f_{\theta_s}$ can model any distribution."

---

### Official Review · Reviewer_fALd · 2023-10-31

**Soundness:** 4 excellent
**Presentation:** 3 good
**Contribution:** 4 excellent
**Rating:** 8
**Confidence:** 3

**Summary:**

This paper presents a theoretically grounded approach based on contrastive learning, SepCLR, to separate salient features from common features given a weak supervision in the form of a target dataset that contains both salient and common features and a background dataset that contains only common features. A series of mutual information based objective functions and regularization terms are presented along with clear motivation and background behind each of the terms, and the approximate loss functions are derived to achieve these objectives. Experimental results on multiple vision and medical benchmarks demonstrate that a good separation of latents is achieved and the method out-performs prior work. Visualizations of retrievals support their experiments.

**Strengths:**

This paper is well-organized and well-written. It covers the necessary background on contrastive analysis, provides the theoretical and intuitive motivations on their various objective functions and puts them in context with prior literature, provide the derivations, and also discuss their limitations well.

The main novelty lies in their information-theoritic formulation of the various objectives and more importantly, exploiting contrastive learning and other prior literature to make the various objectives tractable. Another novelty is that they proposed joint entropy maximization to prevent information leakage between the common and salient latents as opposed to mutual information minimization, as the latter strategy can lose information.

Results in Table 1 and 2 show good latent separation on synthetic vision tasks, and that the proposed approach outperforms prior approaches. In addition, they also reveal that the proposed joint entropy maximization is better than other strategies to prevent information leakage. Table 3, 4, and 5 show similar results on the medical domain and the retrieval visualizations support their findings.

**Weaknesses:**

This paper presents several objectives, but it's unclear which objectives are important and how they work together. This makes it unclear for a practitioner to transfer the results to a different problem. So, I encourage the authors to ablate the different objectives.

Another weakness is that in Table .4, which is perhaps an important real-world application of the proposed approach, the improvements over prior work is not much. This is in contrast with other experiments i.e. Table 1, 2, 3, 5. The reasoning for this is unclear and also not provided.

Nit: MMD abbreviates to maximum mean discrepancy and is referred in the paper as moment matching distance [1].

[1]: A Kernel Two-Sample Test https://jmlr.csail.mit.edu/papers/v13/gretton12a.html

**Questions:**

1. Could you provide an ablation study on the various objectives and their impact?
2. In Table 1, 2, what is the performance on the background dataset? Is the salient latent non-informative as it is supposed to be? Is the common term informative?
3. Is the task switch from classification to regression affecting the proposed approach's effectiveness in Table 4?

---

> ### Author Response · Authors · 2023-11-21
> **Authors response to Question 1**
>
> Q: Could you provide an ablation study on the various objectives and their impact ?
>
> A:  We provide an ablation study in Tables 13 and 14 of Sec. G.4 in the revised manuscript. In the original paper, we investigated the effect of the null Mutual Information constraint by removing the proposed loss (No k-JEM) or by minimizing the Mutual Information estimate (k-MI) rather than the Joint Entropy estimate (see Tab.1 and Tab.2). Here, we propose a further ablation study in Tables 13 and 14 of Sec. G.4 . We report the results of our method when removing all proposed losses one by one. We can observe that each loss is important since, when removing it, we either degrade the capture of salient patterns or we fail to disregard the common features in the salient space.
>
> Table 13: Ablation Study on Digits watermarked on CIFAR-10 (B-ACC).
>
> |                                              | Digits       | Digits         | Objects        | Objects      | $\delta_\text{tot} \downarrow$ |
> |----------------------------------------------|--------------|----------------|----------------|--------------|--------------------------------|
> |                                              | S $\uparrow$ | C $\downarrow$ | S $\downarrow$ | C $\uparrow$ |                                |
> | SepCLR-$\lambda=0$ (no k-JEM)                | 95.6         | 94.4           | 9.0            | 42.0         | 145.8                          |
> | SepCLR-$\beta=0$ (no Infoless reg)           | 96.2         | 11.6           | 10.4           | 71.8         | 34.0                           |
> | SepCLR-$\lambda_S=0$ (no Salient term)       | 93.4         | 42.0           | 18.6           | 40.0         | 90.25                          |
> | SepCLR-$\lambda_C=0$ (no Common term)        | 94.4         | 10.4           | 18.8           | 20.4         | 94.4                           |
> | SepCLR-k-JEM                                 | 96.2         | 11.0           | 10.4           | 73.2         | 32.0                           |
> | Best expected                                | 100.0        | 10.0           | 10.0           | 100.0        | 0.0                            |
>
>
> Table 14: Ablation Study on Digits watermarked on CelebA with accessories (B-ACC).
> |                                              | Hats/Glss    | Hats/Glss      | Sex            | Sex          | $\delta_\text{tot} \downarrow$   |
> |----------------------------------------------|--------------|----------------|----------------|--------------|--------------------------------|
> |                                              | S $\uparrow$ | C $\downarrow$ | S $\downarrow$ | C $\uparrow$ |                                |
> | SepCLR - $\lambda=0$ (no k-JEM)              | 99.03        | 66.68          | 98.48          | 79.48        | 86.65                          |
> | SepCLR - $\beta=0$ (no Infoless reg)         | 99.12        | 53.88          | 68.82          | 77.29        | 46.29                          |
> | SepCLR - $\lambda_S=0$ (no Salient term)     | 77.50        | 87.73          | 53.30          | 77.55        | 85.98                          |
> | SepCLR - $\lambda_C=0$ (no Common term)      | 98.38        | 56.32          | 66.44          | 53.09        | 71.29                          |
> | SepCLR - k-JEM                               | 98.57        | 55.21          | 62.52          | 78.00        | 41.16                          |
> | Best Expected                                | 100.0        | 50.0           | 50.0           | 100.0        | 0.0                            |

---

> ### Author Response · Authors · 2023-11-21
> **Authors response to Question 3 and Weakness 2**
>
> Q: Is the task switch from classification to regression affecting the proposed approach's effectiveness in Table 4 ?
>
> A: Results in Table 4 and 5 are about real-world applications and are defined as multiclass classifications (and not regression). As such, these tasks are more complicated than in CIFAR-10 (Table 1) and CelebA (Table 2) and this could explain the differences in performance. Nevertheless, in Table 4 and 5,  we can see that the proposed method clearly outperforms all other concurrent methods on the subtype classification task using the salient space, which is the most important task, and is on par (or a bit better) in the other tasks (see $\delta_{TOT}$). This means that our method captures more relevant information in the salient space, than the other concurrent works, without degrading the performance in the common space.

---

> ### Author Response · Authors · 2023-11-21
> **Typo**
>
> Nit: MMD abbreviates to maximum mean discrepancy and is referred in the paper as moment matching distance [1].
> [1]: A Kernel Two-Sample Test https://jmlr.csail.mit.edu/papers/v13/gretton12a.html'' \\
>
> A: Thank you for spotting this typo. It has been corrected in the revised version of the manuscript.

---

> ### Author Response · Authors · 2023-11-21
> **Authors response to Question 2**
>
> Q: In Table 1, 2, what is the performance on the background dataset? Is the salient latent non-informative as it is supposed to be? Is the common term informative?
>
> A: We evaluate the ability to linearly predict common attributes only in the common space. In Tables 15 and 16 of the revised manuscript, we can see that the common performances remain good on background samples while the salient space is non-informative as it is supposed to be. We can also notice that, compared to concurrent CA-VAE methods, our method is still the best-performing one in terms of $\delta$, as it better predicts common attributes. These two tables have been added in the revised version of the manuscript in Supplementary Materials in Sec. G.5.
>
> Table 15: Balanced Accuracy results on Digits watermarked on CIFAR-10 on background samples.
>
> |                | Digits       | Digits         | Objects        | Objects      |                     | $\delta \downarrow$ |   |
> |----------------|--------------|----------------|----------------|--------------|---------------------|-------------------|---|
> |                | S $\uparrow$ | C $\downarrow$ | S $\downarrow$ | C $\uparrow$ |                     |
> | MM-cVAE        |    x    |    x      | 18.2           | 32.8         | 75.4                |
> | SepVAE         |   x    |     x     | 20.0           | 34.4         | 75.6                |
> | SepCLR-k-JEM   |   x     |    x      | 28.0           | 74.0         | 44.0       |
> | Best expected  |     x   |   x       | 10.0           | 100.0        | 0.0                 |
>
>
>
> Table 16: Balanced Accuracy results on CelebA with accessories.
>
> |                | Hats vs Glss       | Hats vs Glss         | Sex        | Sex      |                     | $\delta \downarrow$ |   |
> |----------------|--------------|----------------|----------------|--------------|---------------------|-------------------|---|
> |                | S $\uparrow$ | C $\downarrow$ | S $\downarrow$ | C $\uparrow$ |                     |
> | MM-cVAE        |    x    |     x     | 64.27          | 70.48        | 43.79               |
> | SepVAE         |   x     |      x    | 56.42          | 70.19        | 36.23               |
> | SepCLR - k-JEM |     x   |     x     | 64.10          | 86.63        | 27.47      |
> | Best Expected  |   x     |       x   | 50.0           | 100.0        | 0.0                 |

---

### Official Review · Reviewer_Btev · 2023-11-01

**Soundness:** 3 good
**Presentation:** 3 good
**Contribution:** 3 good
**Rating:** 8
**Confidence:** 3

**Summary:**

The paper provides a novel concept of contrastive learning for separating common and silent patterns. The approach utilizes two encoders, one responsible for learning silent representation and one for common. The authors propose the criterion to train the model that is based on the InfoMax principle. Due to the fact the direct optimization of the criterion for the problem is difficult and general for the problem, they propose a set of assumptions that allow to training of the model directly using a gradient-based approach. The approach is evaluated using big number of use cases.

**Strengths:**

- The presentation of the paper is good, and the work is clear and well-written.
- The proposed method is very interesting and sounds good technically. The flow of the proposed solution seems to be accurate. The authors clearly formulate the problem and general criterion given by eq. 1. Further, they decompose each component and propose a well-justified form of the component for given encoders.
- The experiments are well-motivated, and the results seem to confirm the hypothesis stated in this work. It is very beneficial that datasets are from a variety of domains, going beyond standard benchmarks.

**Weaknesses:**

- The model is designed only to model $p(c|\cdot)$ and $p(s|\cdot )$. It would be nice to see some approximation of the distribution over data $p(\cdot|s,c)$. I think that the proposed architecture can be enriched with the decoder that models this probability.
- I am not quite sure if setting the same architecture for the proposed and reference methods is a good approach. In my opinion, the best architecture for each individual method should be used in experiments.
- Only VAE-based methods are used as reference approaches. VAEs are doing the additional jobs, they have decoders and serve as generative models, while SEPCLR is only learning the common and silent representation. It would be nice to see the comparison with the models from similar groups, either SEPCLR as VAE, or reference methods that do not preserve autoencoding properties.

**Questions:**

I would like to ask the authors to respond to the weaknesses section. I will also would like to ask about selecting KDE as a model for this case. Can KDE be replaced with the normalizing flow instead?

---

> ### Author Response · Authors · 2023-11-21
> **Authors response to Weakness 1**
>
> We thank the Reviewer for the feedback. Here is our response.
>
> Q: The model is designed only to model $p(c|.)$ and $p(s|.)$. It would be nice to see some approximation of the distribution over data $p(.|c, s)$. The proposed architecture can be enriched with the decoder that models this probability.
>
> A: This is a good point. Contrastive Analysis, jointly performed with a generative process, enables performing salient or common characteristics swapping, salient attribute generation or deletion, and novel sample generation. Nevertheless, in this paper, we were more interested in capturing semantically expressive representations with a good separation between common and target-specific attributes. Still, we investigated the addition of a decoder, jointly trained with the encoders, to reconstruct the input images (with a Mean Square Error Cost Function) during the optimization process. Intriguingly, we found that it tends to degrade the results (see Supplementary Sec. G.2, Tab. 8), which could be explained by the fact that the reconstruction task tries to conserve unnecessary noisy information in the latent space. However, it could be interesting to include and train a generator or a decoder for generation and interpretability purposes, given frozen representations learned with SepCLR. We leave this exciting perspective as a future work.
>
> In the main text, we propose to clarify the Sec. G.2 by adding: ''Contrastive Analysis, jointly performed with a generative process, enables performing salient or common characteristics swapping, salient attribute generation or deletion, and novel sample generation. Thus, we investigated the addition of a decoder, jointly trained with the encoders, to reconstruct the input images (with a Mean Square Error Cost Function) during the optimization process. Intriguingly, we found that it tends to degrade the results (see Tab. 8), which could be explained by the fact that the reconstruction task tries to conserve unnecessary noisy information in the latent space. However, an interesting perspective could be to include and train a generator or a decoder for generation and interpretability purposes, given frozen representations learned with SepCLR.''

---

> ### Author Response · Authors · 2023-11-21
> **Authors response to Weakness 2**
>
> Q: I am not quite sure if setting the same architecture for the proposed and reference methods is a good approach. In my opinion, the best architecture for each individual method should be used in experiments.'
>
> A: We apologize for the lack of clarity. The sentence "Please note that the same encoder architecture was used for all methods." only refers to Contrastive Analysis Variational Auto-Encoders methods, where the best encoder architecture was actually the same for all CA-VAE in each experiment. Architectures are described in the Supplementary Materials for each experiment (Sec. F). We propose to clarify this sentence by replacing it with "In each experiment, all CA-VAE use the same encoder-decoder architecture, as described in the Supplementary F. The architecture used for SepCLR is also described in Sec. F." Concerning SepCLR, we make use of a ResNet-18 encoder architecture (one for the common encoder and another one for the salient encoder) for CIFAR-MNIST, CelebA , ODIR and CheXpert experiments. This architecture was chosen since it was the top-performing one in all experiments. To clarify this point, we have added a sentence at the end of the Hyperparameters section: "Architectures and hyper-parameters are chosen as the top-performing ones for each experiment."
>
> In Tab 1 et 2, we show the performance of SepVAE and SepCLR on Digits watermarked on CIFAR-10 and CelebA with accessories with different architectures. SepVAE with ResNet-18 performs less better or similarly than the one described in our original paper. Conversely, SepCLR with ResNet-18 performs better. Overall, SepCLR remains largely better than SepVAE and MM-cVAE, the most consistent methods across Contrastive Analysis VAEs. We have added these two Tables (Tab.19 and Tab.20) in Sec. G.7 of the Supplementary.
>
> Concerning the neuro-psychiatric experiment, VAEs and SepCLR have the same convolutional backbone as described in the Supplementary Materials (Sec. F.6), as we obtained the best results for all methods with this architecture.
>
> Table 19: Results of several different encoder architectures on Digits watermarked on CIFAR.
> |                              | Digits       | Digits         | Objects        | Objects      | $\delta_\text{tot}$ $\downarrow$ |
> |------------------------------|--------------|----------------|----------------|--------------|-------------------------|
> |                              | S $\uparrow$ | C $\downarrow$ | S $\downarrow$ | C $\uparrow$ |                         |
> | SepVAE                       | 90.6         | 17.8           | 10.6           | 36.6         | 81.2                    |
> | SepVAE - ResNet 18 encoder   | 90.8         | 23.2           | 10.2           | 34.0         | 88.24                   |
> | SepCLR with SepVAE's encoder | 75.6         | 28.8           | 16.2           | 52.6         | 96.8                    |
> | SepCLR-k-JEM                 | 96.2         | 11.0           | 10.4           | 73.2         | 32.0                    |
> | Best expected                | 100.0        | 10.0           | 10.0           | 100.0        | 0.0                     |
>
>
>
> Table 20: Results of several different encoder architectures on CelebA accessories.
> |                                      | Hats/Glss    | Hats/Glss      | Sex            | Sex          | $\delta_\text{tot} \downarrow$ |
> |--------------------------------------|--------------|----------------|----------------|--------------|--------------------------------|
> |                                      | S $\uparrow$ | C $\downarrow$ | S $\downarrow$ | C $\uparrow$ |                                |
> | SepVAE                               | 84.46        | 65.19          | 60.12          | 59.20        | 81.65                          |
> | SepVAE with ResNet-18 encoder        | 86.13        | 67.47          | 60.04          | 61.93        | 81.45                          |
> | SepCLR - k-JEM with SepVAE's encoder | 97.89        | 60.01          | 51.07          | 70.51        | 42.68                          |
> | SepCLR - k-JEM                       | 98.57        | 55.21          | 62.52          | 78.00        | 41.16                          |
> | Best Expected                        | 100.0        | 50.0           | 50.0           | 100.0        | 0.0                            |

---

> ### Author Response · Authors · 2023-11-21
> **Authors response to Weakness 3**
>
> Q: Only VAE-based methods are used as reference approaches. VAEs are doing the additional jobs. They have decoders and serve as generative models, while SepCLR is only learning the common and silent representation. It would be nice to see the comparison with the models from similar groups, either SepCLR as VAE, or reference methods that do not preserve autoencoding properties.
>
> A: We tried an encoder-decoder architecture for SepCLR by adding a reconstruction term, but it actually degraded the results (see Point 1).
> We propose to compare SepCLR with self-supervised methods that are not based on the encoder-decoder architecture. Since there are no Contrastive Learning methods tailored for Contrastive Analysis, we propose a naive and simple strategy to compare SepCLR with. First, we infer the common features by training SimCLR [A] only on the background dataset (as it should only have common features). Then, we propose to infer the salient space with the SupCon [B] method trained to discriminate the background samples from the target ones. In this way, we should capture only target-specific patterns while discarding common features. Following the suggestion of Reviewer etAa, we also  compare our method with SimCLR trained on both datasets. However, in this case, the common space coincides with the salient one, and therefore it's not possible to perform the separation between common and salient patterns. Results are shown in the Tables 10, 11, 12. We can notice that SepCLR always performs better in terms of $\delta_\text{tot}$. We propose to add these results in the Supplementary Materials, Sec. G.3, Tables 10, 11, 12 of the revised manuscript.
>
> [A] Chen et al, 2021 SimCLR
> [B] Khosla et al, 2021 SupCon
>
> Table 10: Comparison of SepCLR-k-JEM with Contrastive methods on Digits watermarked on CIFAR-1 (B-ACC)
>
> |                     | Digits       | Digits         | Objects        | Objects      | $\delta_\text{tot} \downarrow$ |
> |---------------------|--------------|----------------|----------------|--------------|--------------------------------|
> |                     | S $\uparrow$ | C $\downarrow$ | S $\downarrow$ | C $\uparrow$ |                                |
> | SimCLR on TG and BG | 44.0         | 44.0           | 94.6           | 94.6         | 180.0                          |
> | SimCLR + SupCon     | 41.4         | 51.4           | 19.0           | 50.0         | 159.0                          |
> | SepCLR-k-JEM        | 96.2         | 11.0           | 10.4           | 73.2         | 32.0                           |
> | Best expected       | 100.0        | 10.0           | 10.0           | 100.0        | 0.0                            |
>
>
>
> Table 11: Comparison of SepCLR-k-JEM with Contrastive methods on CelebA accessories (B-ACC).
>
> |                     | Hats/Glss    | Hats/Glss      | Sex            | Sex          | $\delta_\text{tot} \downarrow$ |
> |---------------------|--------------|----------------|----------------|--------------|--------------------------------|
> |                     | S $\uparrow$ | C $\downarrow$ | S $\downarrow$ | C $\uparrow$ |                                |
> | SimCLR on BG and TG | 98.92        | 98.92          | 84.16          | 84.16        | 100.0                          |
> | SimCLR + SupCon     | 97.93        | 82.15          | 59.98          | 80.76        | 63.44                          |
> | SepCLR-k-JEM        | 98.57        | 55.21          | 62.52          | 78.00        | 41.16                |
> | Best Expected       | 100.0        | 50.0           | 50.0           | 100.0        | 0.0                            |
>
>
>
> Table 12: Comparison of SepCLR-k-JEM with Contrastive methods on ODIR dataset (B-ACC).
>
> |                     | Subtype      | Subtype        | Sex            | Sex          | $\delta_\text{tot} \downarrow$ |
> |---------------------|--------------|----------------|----------------|--------------|--------------------------------|
> |                     | S $\uparrow$ | C $\downarrow$ | S $\downarrow$ | C $\uparrow$ |                                |
> | SimCLR on BG and TG | 66.10        | 66.10          | 57.20          | 57.20        | 125.0                          |
> | SimCLR + SupCon     | 68.70        | 57.17          | 51.94          | 58.41        | 107.0                          |
> | SepCLR-k-JEM        | 68.54        | 47.71          | 52.48          | 59.62        | 97.03                 |
> | Best Expected       | 100.0        | 25.0           | 50.0           | 100.0        | 0.0                            |

---

> ### Author Response · Authors · 2023-11-21
> **Authors response to Reviewer's question**
>
> Q: I would like to ask the authors to respond to the weaknesses section. I will also would like to ask about selecting KDE as a model for this case. Can KDE be replaced with the normalizing flow instead ?'
>
> A: We believe that this should be possible, and we are actually working on that, following recent works [A] Kalatzis et al. 2022, [B] Papamakarios et al., 2022, [C] English et al., 2023. However, this would also increase the computational power and time, and it would also complexify the optimization. This is left as future work.
>
>
> [A]: https://arxiv.org/pdf/2106.03500.pdf
>
> [B]: https://jmlr.org/papers/volume22/19-1028/19-1028.pdf
>
> [C]: https://arxiv.org/pdf/2307.14839.pdf

---

### Author Response · Authors · 2023-11-21
**Authors general response**

We thank the Reviewers for their thoughtful and useful feedback. We have provided our response to each of the points raised. Based on the received reviews, we have updated our manuscript (changes highlighted in blue).

---

### Meta-Review · Area_Chair_hwm7 · 2023-12-06

**Metareview:**

This paper proposes to use contrastive representation learning for contrastive analysis of salient vs. common factors between a background dataset (common only) and a target dataset (salient and common). The authors use two encoders (for the salient and common representation spaces), and train the model based on the InfoMax principle. The authors construct a set of training objectives and regularization terms, based on mutual information, and the approach is evaluated on a large set of datasets to demonstrate better separation of common & salient factors compared to baselines. Reviewers find the approach novel and well-motivated, and additionally, some reviewers appreciates that the experiments are across multiple domains. There was also some detailed discussions between authors and reviewers, which addressed many concerns raised. I encourage the authors to incorporate the valuable feedback from reviewers into their paper. One remaining concern raised by some reviewers is that the method is trained on a combination of many losses. While there is an ablation study, the results are not really clear on how all the terms work together (and how much weighting each loss might affect the training). Finally, I recommend the authors improve the presentation and readability of their work - for example, there are some large paragraphs that could be broken down throughout the paper; the experiments are compacted together and it's difficult to understand the goal of each experiment; and figures (ex: Fig 2) could use more spacing.

**Justification For Why Not Higher Score:**

This method is trained with a combination of different losses, but it is not clear how sensitive the method is to hyperparameter tuning, and further analysis of how the losses interact with each other during training would also be helpful. Reviewers also requested additional qualitative results, which authors have promised for the final version of the paper. However, without seeing qualitative results on real datasets, it is difficult to judge the quality and applicability of the method in real-world settings.

**Justification For Why Not Lower Score:**

Reviewers generally agree on the novelty and technical contribution of this work. Additionally, the experiments across a few different domains are promising. The authors were also able to address many reviewer concerns during the rebuttal process.

---

### Decision · Program_Chairs · 2024-01-16

Accept (poster)